Manuscript prepared for Clim. Past
with version 2014/07/29 7.12 Copernicus papers of the LaTeX class copernicus.cls.
Date: 4 May 2017

# Pseudo-proxy tests of the analogue method to reconstruct spatially resolved global temperature during the Common Era

Juan José Gómez-Navarro[1,2,3], Eduardo Zorita[4], Christoph C. Raible[1,2], and Raphael Neukom[2,5]

[1]Climate and Environmental Physics, Physics Institute, University of Bern, Switzerland
[2]Oeschger Centre for Climate Change Research, University of Bern, Switzerland
[3]Department of Physics, University of Murcia, Spain
[4]Institute of Coastal Research, Helmholtz-Zentrum Geesthacht, Geesthacht, Germany
[5]Institute of Geography, University of Bern, Switzerland

*Correspondence to:* J.J Gómez-Navarro (jjgomeznavarro@um.es)

**Abstract.**

This study addresses the possibility to carry out spatially resolved global reconstructions of annual mean temperature using a worldwide network of proxy records and a method based on the search of analogs. Several variants of the method are evaluated, and their performance is analysed. As a test bed for the reconstruction, the PAGES2K proxy database (version 1.9.0) is employed as predictor, the HadCRUT4 dataset is the set of observations used as predictand and target, and a set of simulations from the PMIP3 simulations are used as pool to draw analogs and carry out Pseudo Proxy Experiments (PPE). The performance of the variants of the Analog Method (AM) is evaluated through a series of PPEs in growing complexity, from a perfect-proxy scenario to a realistic one where the pseudo-proxy records are contaminated with noise (white and red) and missing values mimicking the limitations of actual proxies. Additionally, the method is tested by reconstructing the real observed HadCRUT4 temperature based on the calibration of real proxies. The reconstructed fields reproduce the observed decadal temperature variability. From all the tests, we can conclude that the analog pool provided by the PMIP3 ensemble is large enough to reconstruct global annual temperatures during the Common Era. Further, the search of analogs based on a metric that minimises the RMSE in real space outperforms other evaluated metrics, including the search of analogs in the range-reduced space expanded by the leading EOFs. These results show how the AM is able to spatially extrapolate the information of a network of local proxy records to produce a homogeneous gap-free climate field reconstruction with valuable information in areas barely covered by proxies, and make the AM a suitable tool to produce valuable climate field reconstructions for the Common Era.

## 1 Introduction

Climate Field Reconstruction (CFR) methods (Rutherford et al., 2005; Luterbacher et al., 2004; Mann et al., 2008; Smerdon et al., 2010) aim at reconstructing the spatially resolved time evolution of climate fields based on the information contained in a relatively sparse network of proxy archives, which usually encode only local information about past surface climate. The reconstruction of the two dimensional evolution of past near-surface temperature, in contrast to point-wise temperature reconstructions, can provide insights about the physical mechanisms that are responsible for past climate variability and also about the spatial temperature response to external forcing. However, the information about past climate variability is contained in proxy records that archive past environmental conditions at the local scale. To achieve spatially resolved reconstructions, the different proxy records have to be combined in proxy networks to cover wider regions, and additionally some type of method is required to interpolate, and sometimes also to extrapolate, this information and reconstruct complete gridded climate fields. The most widely applied CFR methods make use of the observed spatial co-variability of climate fields to up-scale the scattered information provided by the proxy records to finally obtain a complete gridded reconstruction of particular climate variables. However, this is not the only strategy possible. In this study, we test the performance of a more recent CFR method, the AM, that does not necessarily estimate the spatial climate co-variability from observations but instead combines proxy records and climate simulations to reconstruct the global near surface temperature field.

There are different types of statistical CFR methods. Point-by-point regression (Cook et al., 2004) establishes a series of linear regression models between each grid-cell of a gridded observational data set and several proxy records located in the vicinity of that particular grid-cell. Once this local regression model is calibrated, the local climate is reconstructed based on those few proxy records, repeating this procedure for all grid-cells until the area of interest is covered. Other CFR methods, based on Principal Component Regression (Luterbacher et al., 2004) or Canonical Correlation Analysis (Smerdon et al., 2010) estimate from observations the modes of spatial co-variability of the climate variable and uses the leading modes as predictands in a multivariate regression model, in which all available proxy records are used as predictors. Other methods are based on the Regularized Expectation Maximization algorithm (Rutherford et al., 2005; Mann et al., 2008) originally designed to fill in gaps in panel data. This method also estimates the spatial climate co-variability from observations, although not in the form of spatial modes as Principal Components Regression or Canonical Correlation.

Statistical CFR methods share common features. One of them is that they are usually based on the assumptions of a linear link, which should be stable over time, between variations in the proxy record and variations in the local climate. Another common assumption is that the climate spatial co-variability was in the past the same as it is observed in the current climate. More modern methods, like Bayesian Hierarchical Modelling (BHM) (Tingley and Huybers, 2009; Werner et al., 2013;

Luterbacher et al., 2016), set up a more complex Bayesian statistical model that describes the link between the local climate and the proxy record and the spatio-temporal co-variability of the climate fields. The parameters of this statistical model are estimated by a Bayesian strategy, resulting in a probabilistic reconstruction of past climate conditional on the values attained by the proxy records in each time step in the past. These more flexible methods may describe the link between proxy record and climate variable in more complex ways than just as a linear function and may incorporate previous mechanistic knowledge about the nature of the proxy record. Similarly, the precise form of the statistical model that represents the spatio-temporal co-variability of the climate field is supported by our knowledge of the present climate, and thus is also based, although indirectly, on the observed climate co-variability.

The AM was originally introduced in the 1970s for weather forecasting (Lorenz, 1969). It is however a rather general framework that allows it to be used in different contexts, and in particular it has found application in various areas of paleoclimatology. Overpeck et al. (1985) studied the sensitivity to the choice of different distances, and demonstrated how the method is able to produced good results using pollen data and biological assemblages. Guiot et al. (1989) used it to produce climate reconstruction based on two European pollen records. More recently, the method has been employed in combination to tree rings reconstructions as a mean to fill gaps in the predictor matrix (Nicault et al., 2008; Guiot et al., 2010). Further, Nicault et al. (2008) used a pseudo-proxy approach similar to the one we use through this work to assess the performance of the reconstruction. In this work, we use the AM to produce a CFR reconstruction following an approach similar to Franke et al. (2010) and more recently Gómez-Navarro et al. (2014). Used in this way, the method uses a data-based approach to represent the spatial co-variability of the climate fields. Thereby, instead of estimating those spatial functions from observed data as traditional statistical CFR do, or prescribing functional spatio-temporal co-variability functions as BHM methods do, the AM samples entire fields of a particular climate variable that have been generated in climate model simulations. Those fields that most closely resemble the proxy patterns at a certain time step in the past are selected for the spatially resolved reconstruction. The reconstructed field may be defined as the most similar simulated field, an average of the most similar fields or, in more complex settings, a function of the whole set of most similar fields. In the case of the most simple setting, in which only the most similar field is selected for the reconstruction, the spatial co-variability is automatically ensured, either that from observations or from a state-of-the-art climate model. In other settings, in which the reconstructed field is constructed from several analog fields, the reconstructed spatial co-variability will not exactly match that from observations or from a simulation, but in general it will be reasonably close. This is one of the main advantages of the AM, and can be extended to the reconstruction of other variables that are not represented by the proxy records. Given a time step in the past, once the field most similar to the proxy pattern has been identified, fields of other variables that have been

simultaneously observed (or simulated) can be taken as a reconstruction that is physically consistent with the pattern provided by the proxy data.

The concept of the AM is therefore similar to offline data assimilation techniques that have been applied in the paleoclimate context over the last few years (Bhend et al., 2012; Steiger et al., 2013; Hakim et al., 2016). These methods use a statistical function (typically a Kalman filter) to update

the prior estimation, taken from a simulated climate field, based on the information from the proxy data (e.g. Hakim et al., 2016). The main difference with respect to the AM is therefore that the latter does not update the prior information, but directly uses one sample (or a function of a selection of them) of the model data pool as reconstructed value. As a consequence, the AM does not introduce additional spatial information not originally included within the pool of analogs. This can be seen as

an advantage, since non-climatic noise of individual proxies cannot results in spatial patterns that are inconsistent with model physics. Hence, if the information from an individual proxy is physically inconsistent with the majority of records, this will result in generally larger distance functions, but does not necessarily introduce larger errors in the proximity of the affected record. The AM has been used with different terminologies and settings in several research areas, ranging from the early stages

of numerical weather prediction (Van Den Dool, 1994), through the estimation of future regional climate change (downscaling) (Zorita and von Storch, 1999), to the reconstruction of past surface climate from long instrumental sea-level-pressure records (Schenk and Zorita, 2012).

The AM shares some similarities with the particle filter method put forward by Goosse et al. (2006). The particle-filter method initially runs a set of simulations for a relatively short period of

time, after which they are compared with available local proxy-reconstructions of (usually) annual or seasonal temperature. The simulations that do not resemble the patterns of reconstructed temperature are discarded and those that resemble the reconstructed temperatures are continued forward in time, or are used as a seed of a spin-off simulation ensemble by stochastically perturbing the initial conditions. This method requires, therefore, a large number of simulations and so far has only been

implemented with climate models of reduced computing requirements. Thus, the spatial resolutions and in general the complexity of the model-generated reconstructions are not as sophisticated as full state-of-the art Earth system models. In the AM, in contrast, the analog patterns are searched through the complete simulated time, independently of whether the dates of the identified analogues are close to the date of the proxy-reconstructed temperature pattern. The advantage of this approach is that the

size of simulation ensemble that provides the pool of analogues does not need to be as large as in the particle-filter method. The price paid is, however, that the external forcing of the analogs may be very different from the external forcing of the target pattern. The underlying assumptions are that the spatial covariance of the temperature field is not strongly dependent on the external forcing, or in other words that the shape of the temperature anomaly patterns that are caused by the external forc-

ing are either independent of the nature of the forcing or that internal variability is able to generate anomaly patterns that resemble those caused by the external forcing. If the pool from which the ana-

logues are drawn is large enough, this condition might be fulfilled. This study aims at ascertaining to what extent this underlying assumption holds so that the reconstructions generated by the AM can be trusted.

Since the evolution of the past temperature is not known with certainty, the reconstruction performance of the method is here assessed with the help of virtual experiments conducted with data generated in realistic climate simulations. The assessment is based on pseudo-proxy experiments (PPE) (Mann and Rutherford, 2002; Zorita et al., 2003; von Storch et al., 2004; Rutherford et al., 2005; Smerdon, 2012; Werner et al., 2013; Gómez-Navarro et al., 2014). Paleo-climate simulations

do not generate proxy records, such as tree-ring widths, that may be consistent with the climate evolution simulated by a climate model, but pseudo-proxy records that mimic some of the statistical quantities observed in real proxy records can be generated from climate simulations (Smerdon, 2012). These statistical quantities may in general comprise the link between the proxy record and the local temperature, the statistical persistence of the proxy record, the gaps present in the proxy

record, etc., although in particular PPE only some of these statistical properties are implemented in the pseudo-proxies to test their influence on the final reconstructions. In addition, the network of pseudo-proxies can also be tailored to mimic the network of real proxy sites that are nowadays used to reconstruct the climate of the past few centuries. Once a network of pseudo-proxy records is created within a climate simulation, any reconstruction method can be applied to this network

to pseudo-reconstruct the target variable. The pseudo-reconstructed variable is then compared with the corresponding variable simulated by the climate model, allowing for an assessment of the performance of the method in this ideal circumstances. This is likely an optimistic estimation of the true performance, since real proxies include sources of non-climate variability that are not straight forward to represent with a simple statistical model, and that are likely to cause larger reconstruction

errors.

    The present work is, therefore, not aimed at presenting a climate reconstruction and studying the implications for the history of recent climate change. Such an assessment is beyond the scope of this manuscript and will be addressed in future studies focused on this topic. Instead, the goal of this contribution is to propose and evaluate, mostly with the help of a number of PPEs where the

temporal evolution is borrowed from a climate model run, the performance and major limitations of a CFR method based on the AM. The method aims at producing a reconstruction of the mean annual near-surface air temperature (SAT).

## 2   Data

The study does not critically rely on a particular set of proxy data nor on observations, as the fo-

cus is on the evaluation of the performance method itself. Therefore, the study is mainly based on pseudo-proxy experiments in which the PMIP3 simulations (Braconnot et al., 2012; Taylor et al.,

2012) provide the test bed of the AM. Still, selecting a realistic network that mimics the location of real proxies is crucial to achieve meaningful results that can be then translated to real practice of reconstructions. Nevertheless, the AM has been also tested with observations in the period 1850-2012 Section 5. This requires having both, a network of actual proxies and their previous calibration against observations. Both datasets, as well as the set of simulations used to draw analogs, are briefly described in the following. Further, two different designs of the pseudo experiments are introduced, which are necessary for testing the AM.

## 2.1 Observational dataset

The version 4.3 of the HadCRUT4 dataset (Morice et al., 2012) consists of gridded near-surface air temperature series, calculated as anomalies relative to 1961-1990 mean. It spans the period from 1850 to the present with monthly resolution. The product blends the HadSST3 and CRUTEM4 datasets for sea and land surface temperatures, respectively, and thus provides global coverage with a horizontal resolution of $5°$. The method to produce this dataset generates an ensemble of 100 realisations that allows the characterisation of uncertainty. The ensemble median is used in this study.

An important caveat of HadCRUT4 is the fact that it contains missing values stemming from the lack of meteorological observations in certain barely populated areas. These gaps remain in the final product, since the method applied to the observations does not include data extrapolation. To avoid this drawback, a slightly modified version is considered where missing values have been infilled using a 2-stage GraphEM interpolation (Guillot et al., 2015).

## 2.2 Proxy network

The PAGES 2K Consortium has compiled a global dataset of proxy temperature records. Records were assembled by experts to represent the evolution of temperature over the last 2000 years. Quantitative criteria for record length, resolution and other factors were devolved to select a large dataset that can be culled to address a wide range of research questions (http://www.pages-igbp.org/ini/wg/2k-network/intro). The first version of this dataset, containing 511 proxy records, was used to generate temperature reconstructions for seven continental-scale regions using various reconstruction methods (PAGES2K Consortium, 2013). It has since been updated and expanded to include marine records and additional metadata (PAGES2K Consortimum, 2016). Some records in the 2013 version were excluded because of more stringent selection criteria, which have now been applied more uniformly across regions. We use the version 1.9.0 of this dataset, the predecessor to the slightly revised upcoming version 2.0.0, which will shortly be published (PAGES2K Consortimum, 2016). Thus, the version used herein represents an intermediate snapshot between versions 1 (PAGES2K Consortium, 2013) and 2 (PAGES2K Consortimum, 2016). In total, 682 records are included from 640 terrestrial and ocean locations (Fig. 1). The records belong to 10 types of proxy archives and vary in time

resolution and record duration, being the majority of them tree-rings (61%), with assumed annual resolution. Unfortunately, not all proxies span the full period, as shown in the bottom map in Fig. 1, which depicts the number of years where each proxy does not contain missing values within the period 1-2012. For further details about the database, especially regarding the nature and temporal evolution of data availability, we refer to (PAGES2K Consortimum, 2016). The records with lower time resolution are interpolated to emulate annual resolution, and seasonally-resolved proxies are also processed to remove the annual cycle. This dataset is hereafter referred to as PAGES-FULL.

In addition to this, two slightly different subsets of the dataset are used. The PAGES-SEL includes only those records with native annual resolution, i.e., without interpolation in time, start before 1881, and have less than 1/3 of missing values during the calibration period 1881-1995. This subset contains 514 records. The PAGES-SCREEN is a more restrictive subset, which was screened for a statistically significant correlation with regional temperatures. We use the regional plus FDR (False discovery rate; Ventura et al., 2004) screening from (PAGES2K Consortimum, 2016). This procedure selects only those proxy records with significant ($p < 0.05$) grid cell correlations within a search radius of 2000 km and corrects for FDR. This screening reduces the redundancy of records in areas where they cluster, particularly western North America and the Himalayas (Fig. 1) but also removes records from areas where the proxy density is sparse. This subset consists of just 197 records. Although the influence of using different subsets is addressed in Section 6, most of the analysis hereinafter is based on the PAGES-SEL subset.

### 2.3 Model simulations

The AM method requires a pool of plausible SAT fields to be used for the search of analogs. The size of this pool is crucial, as it needs to cover as many potential climate situations as possible which might have occurred over the Common ERA. To account for this, we use an ensemble of Earth System Model (ESM) simulations, i.e., the simulations of the last millennium within the frame of the PMIP3 initiative (Braconnot et al., 2012). This ensemble is part of the Coupled Model Intercomparison Project fifth phase (Taylor et al., 2012, CMIP5) and is produced with different state-of-the-art models which are also used in the assessment of future climate change (Stocker et al., 2013). The heterogeneity of this ensemble (different parameterizations, components included, etc.) is beneficial for this application, since it allows the analogs to be drawn from a wide range of the spectrum of plausible climate situations, each of them consistent within their own model physics. Although different in some details, all models agree in many fundamental aspects of the temperature evolution ove the Common Era. They are fully coupled ocean-atmosphere general circulation models run with similar spatial resolution. Further, the length of the simulations and the forcings implemented is similar although not entirely consistent across the ensemble (Braconnot et al., 2012; Schmidt et al., 2012; Taylor et al., 2012). In total, 16 simulations are considered from 7 ESMs, resulting in a pool size of 18327 years.

## 3 Methods

### 3.1 Calibration of the reconstructions

The PAGES2K datasets consist of a network of raw, uncalibrated proxies. Thus, using this dataset in the AM method requires a prior calibration of the proxy series to temperature that can be compared to the modelled temperature in the search for analogs. Such calibration is a complex task, since different proxies respond to temperature in a different fashion, and their relationship is contaminated by an unknown and different level of non-climatic noise. Further, different proxies span different periods, which leads to a dataset populated with an amount of missing values that varies through time. These drawbacks require a simple method capable to handle this heterogeneity. It should produce a network of reconstructed temperature records that preserves the largest fraction possible of the climate-related variability. Thereby, a simple univariate linear regression model is employed to deduce a statistical relationship between each proxy and the SAT. The regression is calculated against the closest grid point in the HadCRUT4 dataset during an overlapping period. This fit is performed for each location independently. The regression parameters estimated during the calibration period are then used to obtain a local SAT reconstruction.

The period 1911-1995 is used for the calibration, thereby avoiding the use of the full observational record, and setting some observational data aside for the validation of the reconstruction. Figure 1 shows the correlation between the observations and the raw proxy series during the calibration period. The correlation ranges between -0.56 and 0.63, with 65% of values with an absolute value below 0.2. Although the correlation is modest, it is important to note that these proxies have been carefully selected by experts according to their demonstrated ability to reflect temperature variations with respect to the choice of the calibration period (PAGES2K Consortimum, 2016). Furthermore, these correlation values are robust with respect to the choice of calibration period. Various periods have been tested, including the use of the whole period, and differences are hardly appreciable (not shown).

### 3.2 The AM as reconstruction technique

The AM was first introduced in the 1970s for weather forecasting (Lorenz, 1969). Recently, it has been implemented in a variety of applications in climate research, from hurricane prediction (Sievers et al., 2000; Fraedrich et al., 2003) to downscaling (Zorita and von Storch, 1999) and upscaling Schenk and Zorita (2012) techniques. For the interest of this study, the suitability of this technique to generate CFRs has been recently demonstrated for temperature (Franke et al., 2010) and precipitation (Gómez-Navarro et al., 2014) for Europe. Although the method is explained elsewhere, we briefly outline its key ideas here, following the notation by Gómez-Navarro et al. (2014).

The algorithm requires a set of observations of the multivariate predictand $\mathbf{T}(t)$ available over some time $t$, with concurrent observations of a multivariate predictor $\mathbf{P}(t)$. This predictor shall

be available also at time $t_0$ where no observations of the predictand, the target field variable, are available. The basic idea of the AM is that the value of these unknown $\mathbf{T}(t_0)$ can be approximated by a known value of $\mathbf{T}(t)$ if the predictors $\mathbf{P}(t)$ and $\mathbf{P}(t_0)$ at the target time $t_0$ and a time $t$ in the observation period are sufficiently similar. The set of values $\mathbf{P}(t)$ with the simultaneous information of the predictand $\mathbf{T}(t)$ is generally denoted the pool of potential analogs. Thus, at a given time $t_0$, the method compares $\mathbf{P}(t_0)$ with all the members of the pool by using a metric

$$\Delta(t_i) = \mathrm{dist}(\mathbf{P}(t_0), \mathbf{P}(t_i)) \quad , \forall i \in \mathrm{pool}. \tag{1}$$

The element in the pool with the smallest $\Delta(t_i)$ is called the analog, $\mathbf{P}(\tilde{t}_i)$. Thereby, the reconstructed predictand is defined as the value of the predictand at the analog point in time, which minimises the metric $\mathbf{T}(t_0) = \mathbf{T}(\tilde{t}_i)$.

Although the basic idea is simple, there is still flexibility for tailoring the method to fit different requirements. First, the similarity in (Eq. 1) can be defined in multiple ways by using different metrics, some of which are introduced in the next sections. Additionally, the method can be set to select not just one analog, but identify a set of analogs (e.g. Sievers et al., 2000; Fraedrich et al., 2003). For example, the $N$ closest analogs in the pool (in the sense of the distance given by (Eq. 1)) can be used to produce a weighted average

$$\tilde{\mathbf{T}}(t_0) = \sum_{i=1}^{N} \omega_i \mathbf{T}(\tilde{t}_i) \tag{2}$$

where $\mathbf{T}(\tilde{t}_i)$ denotes the predictand fields of the closest analogs, weighted by $\omega_i$. Again, the weighting can be performed in different ways, e.g., by the distance according to the selected metric or simply by equal weights. Here, we consider only the cases $N = 1$ and $N = 5$, and set all weights to $1/N$, which produces a simple average of analogs. It is important to note that the use of several analogs ($N > 1$) filters out noise, and thus the estimation uncertainty is lower, but has the counterpart of underestimating the time variance variance.

### 3.3 Search for analogs in the real space

The measure of similarity described in Eq. 1 makes use of a distance between two patterns of temperature that has to be evaluated over the network of proxy sites. Note that such distance shall be defined flexible enough to accommodate possible missing values. In this analysis we use two different metrics: correlation and Root Mean Square Error (RMSE).

Correlation is defined as

$$\rho(\mathbf{P}(t_i), \mathbf{P}(t_j)) = \frac{(\mathbf{P}(t_i) - \overline{\mathbf{P}(t_i)}) \cdot (\mathbf{P}(t_j) - \overline{\mathbf{P}(t_j)})}{\sqrt{(\mathbf{P}(t_i) - \overline{\mathbf{P}(t_i)})^2 (\mathbf{P}(t_j) - \overline{\mathbf{P}(t_j)})^2}} \tag{3}$$

where the line over a vector indicates that the mean value across coordinates is computed. RMSE is defined in this notation as

$$\mathrm{RMSE}(\mathbf{P}(t_i), \mathbf{P}(t_j)) = \sqrt{\frac{(\mathbf{P}(t_i) - \mathbf{P}(t_j))^2}{M}} \tag{4}$$

Correlation is a measure of the degree of similarity of two patterns, but does not penalise two fields that may differ by a large constant value. This reduces the ability of the metric to detect changes in the global temperature, as will be shown later. RMSE is a metric that penalises simultaneously the lack of spatial co-variability and differences in mean values. Note that this metric is equivalent, except for a multiplicative constant, to the Euclidean distance between the two vectors $\mathbf{P}(t_i)$ and $\mathbf{P}(t_j)$. Both metrics can be generalised in a natural way to account for missing values in proxy sites. In that case, the summations implicit in the scalar product and in the averages skip those sites, and the constant $M$ has to be decreased accordingly.

### 3.4 Search for analogs in the EOF space

As a variant, the search for analogs can be carried out in the low-dimension space expanded by the leading EOF patterns of the temperature variability. The rationale for using this transformation is that although a temperature field has many dimensions, i.e., as many as grid points, these grid points are strongly interdependent, thus reducing the effective degrees of freedom of the phase space. Further, part of this variability may be spurious and attributable to non climate-related variability in the proxy records, i.e., noise. By decomposing the variability of the field in its main modes, temperature variability can be compressed into a much smaller number of independent variables, each one uncorrelated to the others (von Storch and Zwiers, 2002). The use of EOF techniques to reduce the dimensions for the search of analogs has been explored in previous studies (Zorita and von Storch, 1999; Fernández and Sáenz, 2003).

Here, the leading modes of variability are obtained from the observational dataset HadCRUT4 (where there are no missing values). Once the leading $L$ patterns that explain the desired level of variance (in this study set to $90\%$) are identified, the field can be approximated as the linear combination

$$\mathbf{P}(t) \simeq \sum_{i=1}^{L} \alpha_i(t)\mathbf{EOF}_i \, , \tag{5}$$

where $\mathbf{EOF}_i$ represent the spatial the pattern and $\alpha_i(t)$ the corresponding time series, whose calculation is described below. Thereby, the rank reduction achieved by the change of basis emerges from the fact that the vector $\mathbf{P}(t)$, originally defined through $M$ coordinates in the canonical basis, can be described in the EOF basis by $L$, with $L \ll M$. Once the predictor and predictand at each time step are expressed as linear combination of the observed modes of variability, the AM can be applied directly in this space, with the only modification that the metrics described in Eqs. 3 and 4 have to be applied using the vectors of coordinates $\alpha(t_i)$ and $\alpha(t_j)$, instead of the original fields $\mathbf{P}(t_i))$ and $\mathbf{P}(t_j))$. For the EOF space we focus on a single metric, i.e., RMSE.

Despite its apparent simplicity, the calculation of the vector of coordinates $\alpha_i(t)$ deserves some words of caution when working with fields that contain missing values. In the absence of missing values, the $\mathbf{EOF}_i$ vectors form an orthonormal basis. In this case the $\alpha_i(t)$ vector can be easily

obtained as a matrix multiplication

$$\alpha_i(t) = \mathbf{P}(t) \cdot \mathbf{EOF}_i{}^t \,, \tag{6}$$

where each row is an EOF pattern and the super index $t$ denotes matrix transpose. However, when missing values are present in the vector $\mathbf{P}(t)$, such gaps have to be introduced in the vectors $\mathbf{EOF}_i$. Unfortunately, this modification in the vectors destroys their orthonormality, which implies that the former equation has to be generalised. It can be shown that the general expression is

$$\alpha_i(t) = \mathbf{P}(t) \cdot \mathbf{EOF}_i^t \cdot \mathrm{Cov}(\mathbf{EOF})^{-1} \,, \tag{7}$$

where Cov denotes the spatial covariance matrix of the $\mathbf{EOF}_i$ vectors. In the particular case where they are orthonormal (e.g. when there are no missing values) the covariance matrix is the identity matrix of size $L$, and Eq. (7) becomes equal to Eq. (6).

As a final remark, the coordinates $\alpha_i(t)$ do not contain any missing values, regardless of the gaps present in the original vector $\mathbf{P}(t)$ as missing values are implicitly taken into account in the matrix multiplication used to transform the basis. Thus, all $\alpha_i(t)$ coordinates have the length $L$, independent on the presence of missing values. This simplifies the definition of a distance. Still, the presence of many missing values is undesirable since it increases the uncertainty of the estimation of $\alpha_i(t)$.

### 3.5 Design of pseudo-proxy experiments

As part of the performance evaluation of the AM method, we use PPE. These idealised experiments are profusely used in literature to assess the performance of the CFR reconstructions of temperature (Smerdon, 2012, and references therein) or even precipitation (Gómez-Navarro et al., 2014). The procedure extracts data from a climate simulation at a given set of locations to build a synthetic network of local pseudo-records. This synthetic dataset is used as input for the reconstruction method with the aim to recreate the reconstruction procedure, and then to compare this pseudo-reconstruction with the original simulated field.

The design of PPEs may vary in complexity. The so-called perfect PPEs use the closest grid point to the location of the real proxy to extract a time series of the physical variable of interest. The synthetic reconstructions used as input therefore consist of a simple subset of the original field of the simulation. This is clearly an oversimplification of reality, since actual local reconstructions reproduce only a fraction of the actual climatic signal and include uncertain levels of noise and missing values. A more realistic approach consists of contaminating the climate model series with a certain amount of statistical noise and gaps, so that the starting point of the CFR reconstructions more closely mimics real proxy data.

In this study, we select one of the simulations from the PMIP3 ensemble as a target and to create the pseudo-proxies for the PPE (in particular we use the simulation with the GISS model labeled r1i1p121). We then build the pool of analogs from all other simulations but excluding this simulation,

and reconstruct the target with the AM. Although the results are largely independent on the choice of model, as we indeed demonstrate in Section 4.4, the rationale for this choice is that this simulation is somewhat dissimilar to the other model simulations in that it exhibits lower variability than the other models. This somewhat dissimilarity renders the exercise of reconstructing the target GISS temperature using the other models as pool of analogues more difficult, and therefore it results in a
slightly more strict test.

The network of proxies to base most of our results is the PAGES-SEL network, although other networks are explored in Section 6. All networks of pseudo-proxies consider the real missing values in the PAGES2K network, and thus mimic the reduction of available real proxy records back in time. We employ first perfect PPEs (with no contamination with noise), which allows to assess an upper
limit of the performance of the method and is referred hereafter as NoNoise PPE. In a next step, we consider a more realistic scenario where white noise is added to the series. Other types of statistical noise with different properties can be considered, e.g. red noise produced by an autoregressive process, which allows to simulate the climate memory contained in natural proxy records. Therefore, this study also considers additional tests with red-noise pseudo proxies, prescribing a plausible
time decorrelation of five years. The decorrelation time in actual proxies is not well known, and clearly depends on the nature of the proxy record. Hence, the choice of five years is a pragmatic choice that helps to illustrate the possible effects of red-noise pseudo-proxies without the aim of being overly accurate. In both cases, with red and white noise, the amplitude is set so that it reduces the point-wise correlation with the original series in each proxy location to 0.5. This level of noise,
that corresponds to a signal-to-noise ratio (by standard deviation) of 0.58, is comparable to similar studies (von Storch et al., 2008; Smerdon, 2012; Gómez-Navarro et al., 2014). In this experiment the same missing values present in the PAGES-SEL reconstructions are introduced to mimic a more realistic pseudo-proxy network. This experiment is referred as R0.5 PPE. In a final setup, a set of even more realistic PPEs are carried out in which each pseudo-proxy is constructed with different
amounts of white noise, so that the correlations with the original series equal the correlation values between the real proxy records and observed temperatures, i.e. the values shown in Fig. 1. This is referred as RProxy PPE.

## 4  Evaluation of the AM in PPEs

In this section, only PPEs are used to evaluate the performance of the AM to reconstruct global
annually-resolved temperature. In all cases the full PMIP3 ensemble has been considered by leaving out one simulation, and the proxies location is based on the PAGES-SEL network, as described in Section 3.5.

### 4.1 NoNoise PPE

Figure 2 shows the point-wise correlation maps (calculated for the full reconstructed period) be-
tween the original simulation and the pseudo-reconstructions based on perfect pseudo proxies with
1 and 5 analogs, for a similarity measure based on RMSE, correlation and RMSE in the EOF space,
respectively. All methods tend to produce positive correlations, which is indicative of the ability of
the reconstruction method to recover the original variability based on a limited number of locations.
Still, there are large differences among the different settings. The reconstructions based on the met-
ric of correlation is less reliable than the one based on RMSE. The lack of performance likely stems
from the less demanding criterion of (dis)similarity between two variables that correlation provides,
ignoring shifts in the average fields, and thus focusing just on the spatial co-variability. In this sense,
RMSE presents a compromise, penalising analogs that strongly differ from the target field both in
terms of spatial variability and absolute values. The RMSE similarity is more demanding, and even-
tually the identified analogs are physically closer to the target pattern. The search within the space
spanned by the first EOFs leads to a similar point-wise correlation as in the former case, which is
somewhat expected since the metric is the same, and the phase space, although severely reduced in
terms of number of dimensions, still preserves by construction 90% of the original variance. The
inclusion of more analogs has the effect of increasing the temporal correlation. This effect, also de-
scribed by Gómez-Navarro et al. (2014), is due to the cancellation of errors in the averaging process.
The cancellation of errors has the counterpart of averaging out also a larger part of the reconstructed
variability. Thus, there is a trade-off between temporal accuracy and variance. This is further illus-
trated by Fig. 3, where the ratio of the standard deviations in the reconstruction and the simulation
is presented. Overall, all reconstructions tend to preserve well, and even overestimate, the original
variability. This is a result of the lower variability of the simulation used as target (based on the
GISS model) versus the model ensemble as a whole, and thus re-sampling the pool of analogs tends
to produce larger variability than the target. This overestimation of variability becomes strongly
ameliorated when 5 analogs are used, as expected according to the discussion above.

Spatially, the performance, measured by the point-wise correlation in Fig. 2 is quite homogeneous,
despite the unequal distribution of the proxies and especially despite the smaller number of proxies
in the Southern Hemisphere. Within the North Hemisphere, the area where the reconstruction is less
accurate is clearly the North Atlantic, which stands out across all reconstructions. In this sense, the
EOF-based reconstruction seems more robust, since it does not present the slight negative correla-
tions that appear near the North Atlantic, Caribbean Sea and the Sahara. The areas south of 40°S
show low correlations, which can be clearly associated to the lack of proxies that provide informa-
tion to the reconstruction. Regarding variability, the spatial structure is coherent across methods.
Still, the strong underestimation of variance in all reconstructions in the western North Atlantic is
notable. This underestimation can be directly linked to strong variance in the simulations used as tar-
get (not shown). The consistency of these deficiencies demonstrates how the AM method is always

constrained by the quality of the data used as pool for the analogs search. In this case, the features observed in the target field are not shared across models, which leads to the inability of the method to find suitable analogs that capture certain features.

    Based on the results that emerge from Figs. 2 and 3, the rest of the analysis focusses solely on the reconstructions carried out with the search of analogs in the real space and based on RMSE

similarity (hereafter RMSE-AM), and the one in the EOF space (hereafter EOF-AM). Similarly, only reconstructions using an average of 5 analogs are discussed. However, although not shown, the analysis has been carried out with all combinations of settings, and significant deviations from the results expected from the discussion above are highlighted.

    A very important aspect of this pool of analogs is that it is heterogeneous, since the analogs come

from few very different climate models. Thus, an important question to be addressed is whether there are models that are selected more frequently, and whether there is a strong relationship between the year being reconstructed and the year that correspond to the closest analog. This is shown in Figure 4, where the number of times each model has been selected is shown for each method (panels a and c). All models across the pool are selected at some point in the reconstruction (with the exception of

model number 5, which is the model explicitly excluded for being the target of the PPE). Still, some models are more frequently selected than others. Numbers 1 and 13 are overall the most frequently chosen in both methods, and correspond to the BCC and the IPSL models, respectively. On the other hand, models 15 and 16 are the less frequently chosen models, and correspond to two realisations of the MPI model. It is worth noting that the other simulations with the GISS model (numbers 4 to

11) are not selected more frequently than the rest of models, despite being simulations of the same model as the target. This is indicative of the ability of the search algorithm to identify similarities in the spatial patterns regardless of particular model features, and supports the robustness of the reconstructed fields with respect to the biases present in some models. Black thin lines denote the occurrence of severe volcanic activity, and are aimed at facilitating the identification of relation-

ships between this external forcing and year selection. It turns out however that the method selects analogs independently from this factor. Similarly, there is no strong one-to-one relationship between the simulated and reconstructed years, i.e. simulated modern (or earlier) years are not necessarily selected to reconstruct recent (or earlier) years (see scatter dots in panels b and d). This is indicative of the sufficiently large amount of variability contained in the pool, which thanks to the amount of

internal variability provided by the various simulations, is able to provide analogs independently of the model year. The only signal of a temporal link between the targets and their analogs appears as a clustering of modern simulated years that are used as analogs for years within the 20th century (see the clustering of dots in the top right corners in panels b and d). This is attributable to the effect of recent warming of the industrial period, i.e. warm years appear more frequently, and they are

preferably found during the last centuries of the pool of simulations.

## 4.2 R0.5 PPE

This section explores the performance loss when noisy pseudo proxies are used to mimic the effect of non-climate related variability of real proxy data. As outlined above, the noise consists of additive white noise and the introduction of missing values that mimic the temporal distribution of missing values present in the PAGES-SEL network. Note that, for the sake of brevity, the analysis hereafter is limited to the RMSE-AM and EOF-AM methods for analogs search, although the other methods have been explored and the results are consistent with the former section: i.e. the RMSE metric outperforms correlation as measure of distance between analogs. Similarly, only the reconstruction obtained as an average for the 5 best analogs is discussed, since the 1- and 5-analog versions differ in the bias-variance trade-off described in the perfect scenario context in the previous section.

The performance of the reconstructions with these more realistic PPEs is illustrated in Fig. 5. The top row depicts the correlation between the original simulation and the reconstructions based on realistic PPE contaminated with noise and populated with missing values. The correlation is generally lower than in the case of perfect pseudo-proxies, indicating the reduced performance of the reconstruction method in this scenario. This is expected since the quality of the pseudo-proxies has been considerably degraded in this PPE. However, the decrease in the correlation is remarkably small, from 0.35 to 0.28 and from 0.39 to 0.24 on average and for the RMSE and EOF methods, respectively. In particular, the spatial structure of the correlation maps hardly changes with respect to perfect PPE, being the spatial correlation between the perfect and noisy cases 0.94 and 0.95 for RMSE and EOF, respectively. The modest impact of the addition of a strong component of noise is attributable to the use of an extensive network of proxies: the information contained in the network is to a great extent redundant and represents the same climate signal, which implies that the degradation of the information at a given location can be to a great extent recovered by the reconstruction method through the use of nearby information and by the spatial coherence of the climate field. This recovery of degraded information gives confidence about the CFR methods in general, and in the AM in particular, and suggests that the use of a large network of independent proxies can overcome, to a certain extent, the problems derived from the use of noisy local reconstructions. The two maps in the lower row depict the ratio of standard deviation in the reconstruction and the simulation in logarithmic scale. Both figures are hardly distinguishable (spatial correlation 0.97 and averaged bias of -0.02 ), and coherently point out how the reconstruction recovers about 80% of the original variance independently from the particular method (the logarithm of the ratio averages -0.1 and -0.8 for RMSE and EOF, respectively). The loss of variance with respect to the NoNoise PPE is particularly strong in the western North Atlantic. This underestimation of variance disappears and even becomes an overestimation of variance when just 1 analog is considered (not shown). However, this variant of the method presents lower temporal correlation (not shown), as the correlation-variance trade-off is always present across experiments.

The results obtained with the experiments where red instead of white noise is added to the original series resemble those shown in Fig. 5, and are not shown due to the great similarity with the figures corresponding to white noise. All metrics evaluated indicate that the performance of the reconstruc-
tion is indistinguishable when either white or red noise is considered. Therefore, the presence of memory in proxies seems to play a secondary role in the performance of the AM, and does not degrade noticeably the output of the reconstruction. Note that this result agrees with previous findings in similar studies but aimed at the reconstruction of precipitation (Gómez-Navarro et al., 2014). The effect of red-noise pseudo proxies has been tested in previous studies in the context of regression-
based methods and the Composite plus Scaling method (von Storch et al., 2008), where it was found that, in the case of regression methods, red-noise pseudo-proxies lead to a stronger underestimation of past variability than white-noise pseudo-proxies. However, the influence on other measures of skill that do not rely on the amplitude of variations, like correlation, has not been so far investigated. It is therefore, reassuring that the AM does not lead to either an additional reduction of past
variations or to a loss of correlation skill.

### 4.3 RProxy PPE

Figure 6 depicts the same results as Fig. 5 but for the more realistic PPE, which consists of reducing the correlation by adding white noise in an amount that mimic the values observed in the calibration. The decrease in the correlation compared to a situation with spatially homogeneous noise is apparent
(note the different scale for correlation). The inclusion of more realistic values of correlation severely reduces the ability of the AM method to reconstruct the original simulation. The correlation between the pseudo-reconstruction and the target is especially reduced in the tropics and North America, locations where the skill obtained in more simple PPEs is very remarkable, and perhaps overestimated under the light of this analysis. There are however areas where the correlation is still well preserved,
such as in Europe, central Asia and the western Pacific. A striking finding with respect to the former case is the large difference between the RMSE-AM and EOF-AM methods. Although both methods deal with the same amount of uncertainty, the former clearly outperforms the latter regarding its ability to reproduce the temporal evolution in the target, despite the addition of noise and missing values. Still, the spatial structure of correlation is very similar in the RMSE-AM variant, and
in particular the method remains able to deliver performance in regions with poor proxy coverage. Regarding the preservation of variance, both methods exhibit the same underestimation of variance, which stems from the averaging over 5 analogs, and is absent in both cases when only one analog is used to reconstruct (not shown). Thus, both methods behave similarly regarding the replication of variance.
Based on the results of these PPEs, we conclude that the RMSE-AM method is overall the most reliable, since its performance is more robust across the experiments and analyses we have carried out.

### 4.4 Other simulations as targets

All PPE analysed so far are based on the use of a single model as target. This section explores the sensitivity of the results to the use of the simulations MPI-ESM-P r1i1p1 or CCSM4 r1i1p1 as targets, instead of the GISS r1i1p121. Left column in Figure 7 shows the correlation between the target SAT and the pseudo-reconstructed SAT for three models: GISS (which is the model discussed so far), MPI-ESM-P and CCSM4, in a case where the PPE are designed with red noise as described in Section 4.2. Middle column depicts the ratio of standard deviation of the reconstruction and the target, whereas right column shows RMSE to illustrate other performance metrics than simply correlation and demonstrate how it supports the same conclusions. We focus the discussion on the comparison between GISS and MPI-ESM-P, as the one corresponding to CCSM4 is very similar and therefore omitted. The skill of the pseudo-reconstruction are qualitatively very similar, although there are some regional differences which, however, do not modify the main picture derived from the previous sections. The correlation pattern in the MPI-ESM-P case is very similar to that obtain in the GISS case, with high values of the correlations in the Northern Hemisphere and lower values in the Southern Hemisphere. Both cases also display relatively lower correlations in the central North Atlantic and Central Pacific. The correlations are low in the Southern Ocean, possibly due to the very sparse proxy network here. The patterns of RMSE (right column) is also similar in both cases. The RMSE tends to be higher in the GISS case, confirming our initial assumption that the variability of the GISS model stands slightly out of the ensemble of models, though not dramatically. The RMSE is higher in the polar regions, where it may attain values of the order of 2-3K, and rather uniform and lower values around 0.5 K in the rest of the globe. There is a remarkable difference between both cases in the Western North Atlantic, where the GISS case displays rather large values of the RMSE that are not seen in the MPI-ESM-P case, for which there is no clear explanation at this point. Regarding the preservation of variance (see middle column in Fig. 7), there are small regional deviations which seem model-dependent, although the main picture that stands out in all the three cases is that the reconstruction using 5 analogs leads to a slight but generalised loss of variance. Therefore, the main conclusion we can draw from the analysis above is that the choice of simulation as SAT target does not largely affect the performance of the AM in reconstructing global SAT, and the conclusions drawn from the analysis of the GISS model used as target can be safely extended to other models.

### 5 Reconstruction of the observational period

In this section, the ability of the reconstruction method is explored using real proxies to reconstruct the observed temperature field in the period 1850-2012. For this, a selection of the PAGES-SEL network during the period 1850-2000 is extracted and calibrated during the 1911-1995 period against the infilled HadCRUT4 observational dataset in the way described in the Section 3. The series ob-

tained after calibration are used as input for the RMSE-AM and EOF-AM variants of the AM, and the output is compared to the original observations, with the aim of establishing the performance of the reconstruction.

Figure 8 depicts the results of the comparison between the reconstructed and observed series of SAT, and is the counterpart to Figs. 5 and 6 with actual proxies instead of PPE. Note however that correlations in this figure are not fully comparable to the formers, as they have been calculated over different periods (in the formers, the full 2000-year period is used). As the number of proxies varies through time, the skill obtained is not directly comparable, but somewhat overestimated by the availability of proxies in more recent periods. As before, the results focuses on the RMSE and EOF methods, and when 5 analogs are chosen to obtain the reconstruction. Regardless of the particular method used in the search of analogs, and despite of being a favourable test due to the larger amount of available proxies in the period considered for the calculation, the correlation maps between the reconstruction versus the target (top row) exhibit lower values than both with perfect PPEs and with noisy pseudo-proxies with spatially homogeneous noise (Figs. 2 and 5, respectively). This lower temporal correlation may be due to two reasons. One is that the level of noise employed in the first realistic PPE, inspired by its application in similar studies (von Storch et al., 2008; Smerdon, 2012; Gómez-Navarro et al., 2014), is an underestimation. Indeed, the point-wise correlations between the observed temperature and the proxies during the calibration period ranges between -0.56 and 0.63, with an average of 0.06, which would suggest a higher level of noise in the real world than in the PPE. However, a second reason could originate in a deficient simulation of the typical temperature patterns found in the real world. These low correlations impose an upper limit to the temporal evolution that the calibrated series are able to represent. This can be seen more clearly when comparing Figs. 6 and 8, where especially the RMSE-AM method exhibits very similar spatial pattern and values (again, recall that the PPE is again in disadvantage as correlations in Fig. 6 are calculated for the whole Common Era, including early periods more densely populated with missing values). Note that these figures correspond to actually very different datasets (a PPE versus a real reconstruction of an observational dataset), although by construction of the PPE they have in common the spatial proxy network and the correlation between the proxy and the corresponding local SAT series during the instrumental period.

The reconstructions of the temperature in the observational period produce overall positive correlations with the real temperatures, which match fairly well the values obtained with noisy PPE with spatially varying noise levels, especially the RMSE-AM, and depending on the location reach values above 0.5. The distribution of point-wise correlation is affected by the location of the proxies, and seems to be slightly sensitive to the method employed, especially where the point-wise correlation is not supported by the existence of nearby proxies. Thereby, both methods produce reconstructions that exhibit better performance over Europe, north Canada, eastern Asia or Tasmania. However, RMSE shows locations where the reconstruction leads to remarkable performance despite the low

number of proxies located nearby, such as Western Sahara or the Southern Indian Sea, whereas these spots of remarkable correlation cannot be identified in the EOF reconstruction. Conversely, the use of the RMSE similarity leads to negative correlation in South America and near Antarctica, which are missing in the EOF reconstruction. Regarding the preservation of variance (bottom row), both methods underestimate the variance, as expected to some extent when using an average of 5 analogs. In this sense, the RMSE method clearly outperforms the EOF-based method, which unlike the former strongly underestimates variance in nearly all locations. A noticeable agreement between both methods is the consistent underestimation of variance in the Arctic. This may result from the lower variance in the pool of analogs in this region. All models consistently exhibit lower variance in the Arctic compared to observations (not shown), which leads to systematic variance underestimation and provides an example of unavoidable bottleneck of the AM. It is however worth noting that an alternative or complementary explanation for the differences in variability between observations and simulations in the Arctic regions could be in caveats in the former. This is due to the fact that as outlined in the dataset decryption above, observations in the high Arctic are not real, but infilled using extrapolation techniques which might introduce variance overestimation.

## 6   The role of spatial distribution of proxy sites

The reconstruction performance may also depend on the proxy network used. Therefore we assess the impact of slightly different proxy networks on the reconstruction, using the PAGES-SEL, -FULL and -SCREEN networks described above. The observational period serves as an example.

The correlation maps between the observations in the period 1850-2000 and the different RMSE-AM reconstructions based on these networks are shown in Fig. 9, where also the slightly different distribution of the proxies is shown. Using the original PAGES-FULL network generally improves the point-wise correlation of the reconstruction compared to the PAGES-SEL case (recall that this network contains 682 instead of 514 records). This is especially so in equatorial and sparsely covered areas, indicating that the addition of few records, even when they do not provide real annual resolution or when they contain significant amounts of missing values, can have noticeable positive effects on the reconstruction. A striking result is that the PAGES-SCREEN network provides remarkable performance, despite that it just contains 197 records. This suggests that the accumulation of redundant proxies in certain areas, such as North America or China, may have a counterproductive effect in the reconstruction performance. This is a somewhat counter-intuitive result, since the screening of the network produces a reduction of the available information. However our results indicate that the performance is to a large extent preserved, probably because the screened network contains fewer proxies which exhibit low correlations with the instrumental temperature. The combination of the latter two results support the argument that the best possible network would ideally have a global but also a very homogeneous coverage, making the total number of records of secondary importance.

Figure 10 shows the temporal evolution of the globally averaged SAT in the HadCRUT4 dataset
and the RMSE-AM reconstructions with 1 and 5 analogs using each of the proxy networks described
previously. This figure additionally illustrates the reconstruction performance, and is complementary
to the correlation maps discussed so far. All time series reproduce remarkably well the global warm-
ing captured by observations, including the short cooling period during the 60's. The differences
between different setting of the method are minor, and does not affect this general good agreement,
indicating that the long-term variability can be reproduced with confidence regardless of the network
used to reconstruct the climate variability.

## 7   On the estimation of reconstruction uncertainties

The reconstruction of past climate should include an estimation of the reconstruction uncertainty
that sets the validity of that estimation. Such uncertainty stems in general from different sources,
and often some sources of uncertainty can be better estimated than others. This is the case for the
AM, as briefly explained in this section. It is important to note that the estimation of reconstruction
uncertainty requires hypothesising an underlying theoretical framework for the method. For instance,
an underlying assumption in all reconstructions of past climates is that the proxy records still reflect
the environmental conditions in the same way as they do in the present climate. If this requirement
is not fulfilled, the estimated uncertainty is an unrealistic estimate. As an illustration, let us consider
the well known case of a simple univariate regression model (see for instance von Storch and Zwiers,
2002).

$$T = T_m + (P - P_m)\alpha + \epsilon \tag{8}$$

where $T$ and $P$ denote temperature and proxy, respectively; $T_m$ and $P_m$ denote their mean values,
$\alpha$ is the regression coefficient, and $\epsilon$ is the error term. The uncertainty in the estimation of $T$ given
P has two main sources. One is related to the amplitude of the unresolved variance, given by the
standard deviation of $\epsilon$. However, the other main source is the uncertainty in the estimation of $\alpha$,
let us denote it as $\delta(\alpha)$. As can be demonstrated within the linear regression theory, this second
contribution is approximately proportional to the product $(P - P_m)\delta(\alpha)$. Therefore, for values of $P$
in the middle of the range of the predictor, the main contribution is the amplitude of $\epsilon$, whereas for
values of $P$ far away from $P_m$, the main contribution becomes $(P - P_m)\delta(\alpha)$.

In a similar way, in the application of the AM there are two main contributions. One would be
the amplitude of the error term, i.e. the deviations between the actual and predicted $T$, assuming
that the model analogue is perfect. This contribution is analogous to the unresolved variance, i.e.
the variability of $T$ at a certain point that cannot be solely determined by the given temperatures
at the proxy locations. A second contribution to uncertainty is the identification of the analogue
itself. Unfortunately, the situation in the AM is more complex than in the case of simple univariate

regression. For target patterns where good analogues can be easily be found, this contribution will
be very small. In general, and since we use a large pool for the analogue search, it can be assumed
that for proxy patterns that are 'around the mean', the AM is generally able to find good analogues
within the pool. However, for proxy patterns well beyond the range of the pool, where no good
analogues can be found, the uncertainty cannot be easily quantified. The reason for this is that such an
estimation would require an analytical model, being the counterpart of the regression model outlined
above. Unfortunately such frame model, able to carry out some sort of 'analog extrapolation model',
which would allow to estimate a range of the predicted variable in ranges where no good analog of
the predictor exists, has not been developed yet. Therefore, for targets well beyond the analogue pool,
this contribution to uncertainty would be the largest, although unknown. Note that this situation is,
to some extent, similar to pollen-based reconstructions using the analogue method (Overpeck et al.,
1985). When the pollen record shows a pattern that is not present in the current pollen distribution,
the climate reconstruction and its uncertainty are virtually impossible to estimate. In this regard, new
mathematical developments are required to settle this issue.

Under the light of the former discussion, in this manuscript we have estimated just the uncertainty
arising from one of the two contributions discussed above, i.e. the variability of $T$ at a certain point
that cannot be solely determined by the given temperatures at the proxy locations. To do so, we do
opt by computing the standard deviations of the residuals (reconstructions minus target). For this
computation, we try to mimic the situation that researchers face in real reconstructions, where the
observed temperature field over a reference period would be known, so that the residuals (deviations
between observations and reconstructions) and its standard deviation can be computed. To simulate
as closely as possible this situation, we compute the standard deviation of the differences using the
1850-2005 period, instead of the whole GISS r1i1p1 simulation.

In order to gain insight on the variability of the error attributable to the variable number of missing
values, we have computed this contribution to the uncertainty for two situations, both within the
main pseudo-reconstructions using white-noise pseudo-proxies with a uniform correlation between
the pseudo-proxy and the local temperature of 0.5 and considering 5 analogues (this is, the PPE setup
discussed in Section 4.2). The first case is the best-case scenario, i.e. we use the proxy records of the
PAGES-SEL network available in the year 1949, where no record has missing values. In the second
case, we use the proxy network representing the year 1500, i.e. selecting only the 257 proxies with no
missing values in this year, to illustrate changes in uncertainties back in time. The results are shown
in left column of Fig. 11, and show that the uncertainties are larger in the polar regions, and are in
the order of 1-2K, being smaller in the tropical regions. This is reasonable since in the polar regions
the spatial correlation of temperature tends to be larger and therefore the temperature at the proxy
locations is less capable of determining the temperature in other locations. Further, the variability is
larger in the arctic regions, which inflates the error in this region. This can be appreciated in the right
column of Fig. 11, which shows the same error, but normalised dividing by the standard deviation

of the in the target. Quite remarkably, the number of proxies has little influence in the intensity and distribution of errors. This is in good concordance with the results discussed in Section 6, and once again demonstrates the secondary role of the absolute number of proxies, as a growing number of proxies sometimes increases redundancy without providing independent source of insight.

## 8  Conclusions

This study presents a framework to carry out global CFRs using the AM based on a pool of the PMIP3 ensemble simulations (Taylor et al., 2012). Although the application of the method has been previously employed to carry out European reconstructions of temperature (Franke et al., 2010) and precipitation (Gómez-Navarro et al., 2014), the validity of this method to accomplish a global temperature field reconstruction has not been addressed so far. This is a relevant test, since the large dimensionality of the problem poses concerns about the suitability of available simulations to provide a large-enough pool of situations from which to draw analogs. This study is also novel in being one of the first analysis that benefit from the PAGES2K proxy network (PAGES2K Consortimum, 2016). In this sense, this work takes advantage of the most recent developments in both the climate model and reconstruction communities (PAGES 2k-PMIP3 group, 2015), and represents an example of the power of exercises blending both approaches to gain insight in climate variability within the Common Era.

A number of variations of the method are presented here, since the AM critically depends on the metric used to identify analogs (normally a distance measure between the analog and the target). Testing different metrics shows that the RMSE, which is equivalent to the Euclidean distance, is more suitable than correlation since it penalises deviations in global averages. The search of analogs in the real space, as well as the one expanded by the leading EOFs that explain 90% of the total variance has been explored. Although the EOF version is in principle better suited for the search of analogs due to the reduction of dimensionality of the problem, our results indicate that the search in the real space provides the best results with a consistent performance across the various tests carried out. Further it has the added value of slightly lower computational cost.

Regardless of the metric used and the nature of the reconstruction (real reconstruction or PPE), the method draws analogs without clear preferences for any model in particular. Indeed, when the GISS model is used to perform PPE, the rest of the GISS simulations are not selected preferably over the rest of the ensemble. This indicates that the method draws analogs according to climate situations, rather than systematic biases of a particular model, and thus provides confidence in the method. Further, the results indicate that the inclusion of a large number of simulations from structurally different models has beneficial effects on the quality of the final reconstruction. Further, the PPE results are barely sensitive to the choice of the target, which indicates that the performance obtained through PPE is a robust estimate of the performance of the AM.

The inclusion of a spatially constant amount of noise in the more realistic pseudo-reconstructions does not dramatically affect the CFR performance, supporting the robustness of the method and the ability of the network of proxies to retain the variability of the global man temperature, in spite of local noise. In particular, there is no difference in the performance between the PPE when either white or red noise with a decorrelation time of five years is used. This indicates that the AM is not sensitive to the presence of memory in the local proxies. Still, there is a large difference in the performance obtained with actual proxies and that achieved in PPEs with degraded pseudo-proxies. This difference suggest that the amount of noise might have been underestimated in previous studies based on PPEs (e.g. von Storch et al., 2008; Gómez-Navarro et al., 2014), and lower signal-to-noise ratio shall be employed in realistic PPEs. This is confirmed by our analysis through a more realistic PPE configuration, where the level of noise depends on the proxy site to mimic the one derived from the calibration of real proxies.

Many statistical climate reconstruction methods tend to underestimate climate variability, especially those based on linear methods. The AM is an exception, since the variability of the reconstruction is provided by that of the pool of analogs. Although this might be seen as an advantage, it has the problem that systematic biases in the pool are transferred to the reconstruction. This is particularly the case with the PMIP3 ensemble, which exhibits a reduced variability in the Arctic compared to the infilled observations that might become a prominent drawback in all reconstructions evaluated here. The AM can be adjusted by varying the number of proxies used to draw an analog. If more than one analog are selected and averaged to generate the analog, the correlation is increased, but it has the counterpart of reducing variability. This bias-variance trade-off is not unexpected, as it is a common phenomenon that appears recurrently in all branches of statistics.

The sensitivity of the CFR to various slightly different versions of the proxy network has also been evaluated. The skill of the reconstruction does not critically depend on the total number of records. Instead, it is more strongly affected by their spatial distribution. In this sense, including redundant proxies that cluster in some areas does not always have a beneficial effect, since they do not provide new information, but may bias the search of analogs towards those areas at the coast of producing less accurate reconstructions in areas less well covered by proxies.

The AM produces climate reconstructions which are clearly not free of uncertainties and errors. However a full treatment and characterisation of such errors is not tackled in this study, as such as assessment would require new mathematical development which are beyond the scope of this article. Still, we investigate a part of such uncertainty, namely the one attributable to the unresolved variance. We characterise it by computing the standard deviation of the residuals using two different networks of pseudo-proxies, and demonstrate how such uncertainty is bounded by 1-2 K in the polar regions, being smaller in the tropical ones.

Finally, we would like to remark that as the performance of the AM has been evaluated in this paper mostly through PPE, and although we have tried to mimic the limitations of actual data, we

note that our estimation of skill can be optimistic, especially in the Southern Hemisphere. This is so due to the fact that reconstructions show less homogeneity back through time than the models are used in this study. For instance, it has been reported that the co-variability between both hemispheres is larger in models than in current reconstructions (Neukom et al., 2014; PAGES 2k-PMIP3 group, 2015).

We conclude that the AM is a useful tool able to yield skillful results in CFRs of past climate. It has particular features compared to more commonly used CFR techniques, e.g. it is a non-linear method that does not require the calibration of an underlying statistical model. Thus, the method may complement more traditional approaches providing additional insight about past climate variability, and allowing to assess the robustness and weaknesses of other methods.

*Acknowledgements.* This is a contribution to the PAGES 2K Network. Researchers of the PAGES 2K Consortium are thanked for creating and releasing the database of proxy data and metadata. Julien Emile-Geay and Nick McKay provided the data files of the PAGES 2K database and the PAGES-SCREEN dataset used herein. Darrell Kaufmann provided inputs on the data section.

We acknowledge the World Climate Research Programme Working Group on Coupled Modelling, which is responsible for CMIP, and we thank all the climate modeling groups for producing and making available their model output.

This work was funded by the Oeschger Centre for Climate Change Research and the Mobiliar lab for climate risks and natural hazards (Mobilab). JJGN acknowledges the funding provided through the contract for the return of experienced researches, resolution R-735/2015 of the University of Murcia and the CARM for the funding provided through the Seneca Foundation (project 20022/SF/16). CR acknowledges support from the Swiss National Science Foundation. RN is supported by the Swiss NSF grant PZ00P2_154802.

The authors would like to thank the reviewers for the time devoted to carefully read the manuscript and provide very useful insight.

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

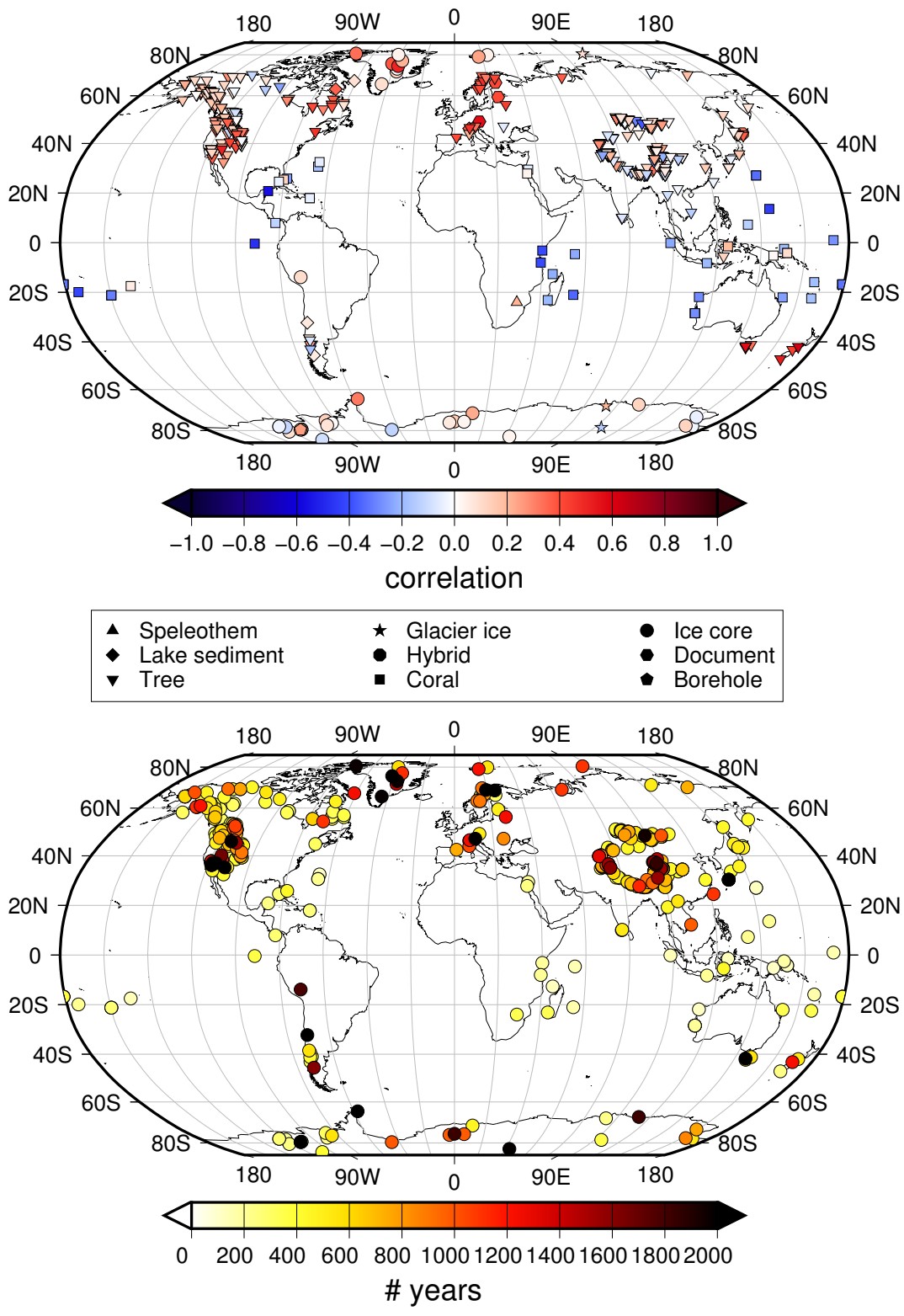

**Figure 1.** Top: point-wise correlation between the raw proxy series in the PAGES-SEL network and the SAT in the infilled HadCRUT4 dataset during the period 1911-1995. Each type of proxy is indicated with a different symbol. Bottom: number of years in which each record contains valid data, i.e. lighter colours indicate shorter records.

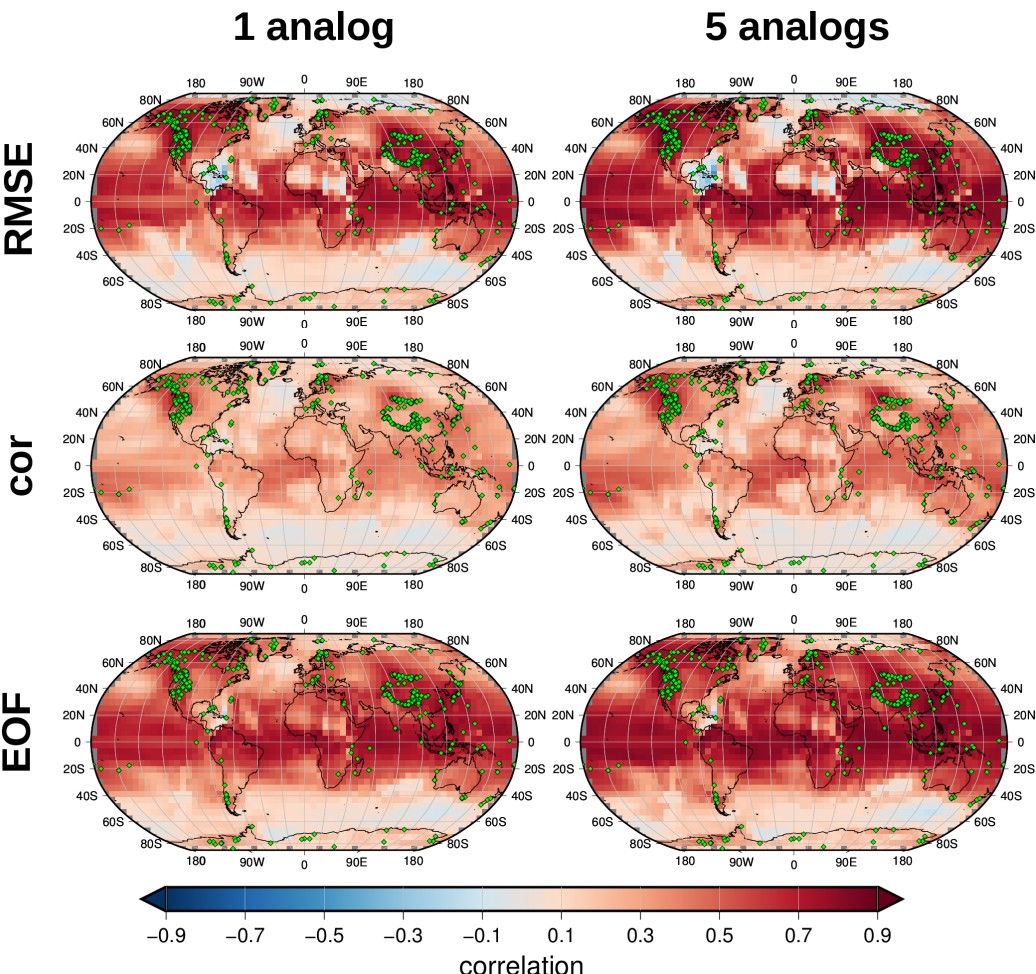

**Figure 2.** Point-wise correlation (calculated for the whole reconstructed period) between the original simulation and a reconstruction based on perfect pseudo-proxies. The maps show the results when three different metrics are used for the search of analogs (by rows), as well as when different numbers of analogs are combined to draw the reconstruction (by columns). Green diamonds indicate the location of the pseudproxies employed, based on the PAGES-SEL network.

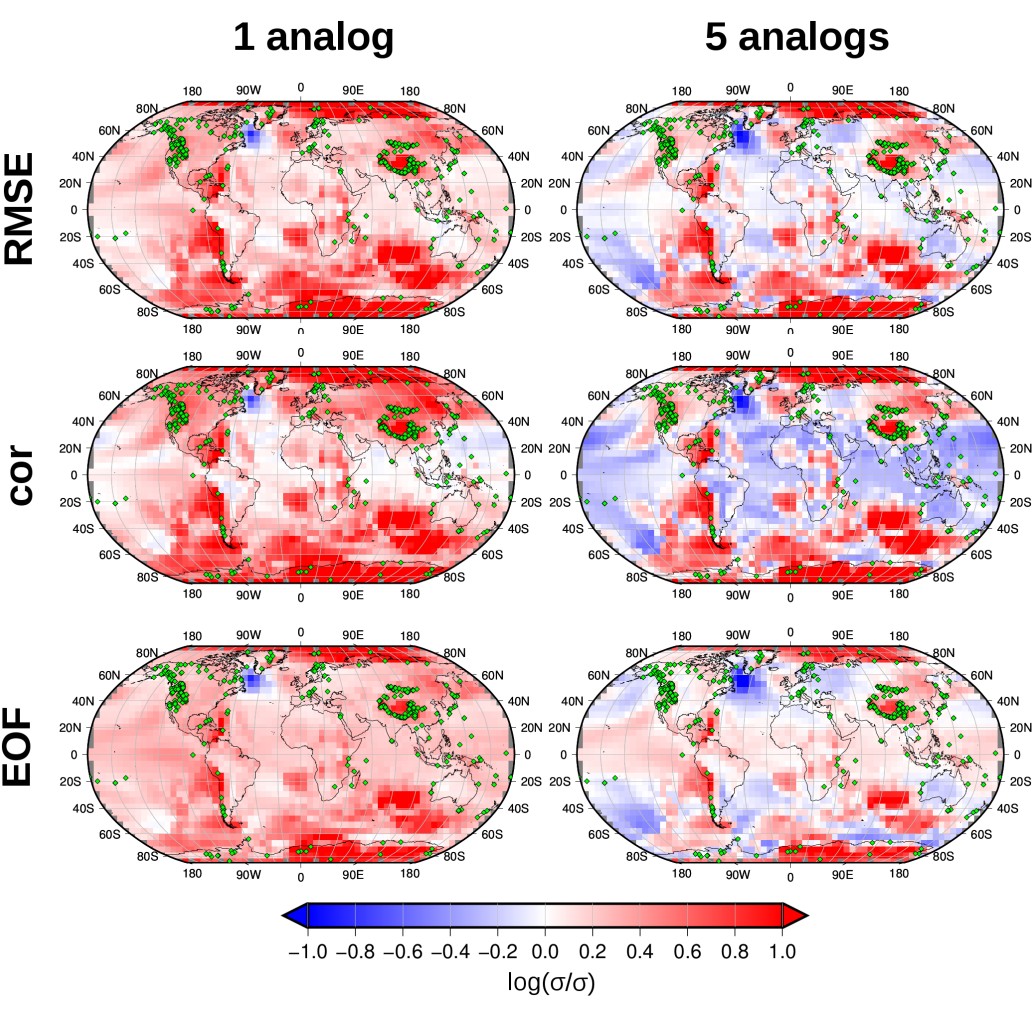

**Figure 3.** As Fig. 2, but for the logarithm of the ratio of the standard deviation of the reconstruction and the original simulation. Red (blue) shading depicts areas where the reconstruction overestimates (underestimates) variability.

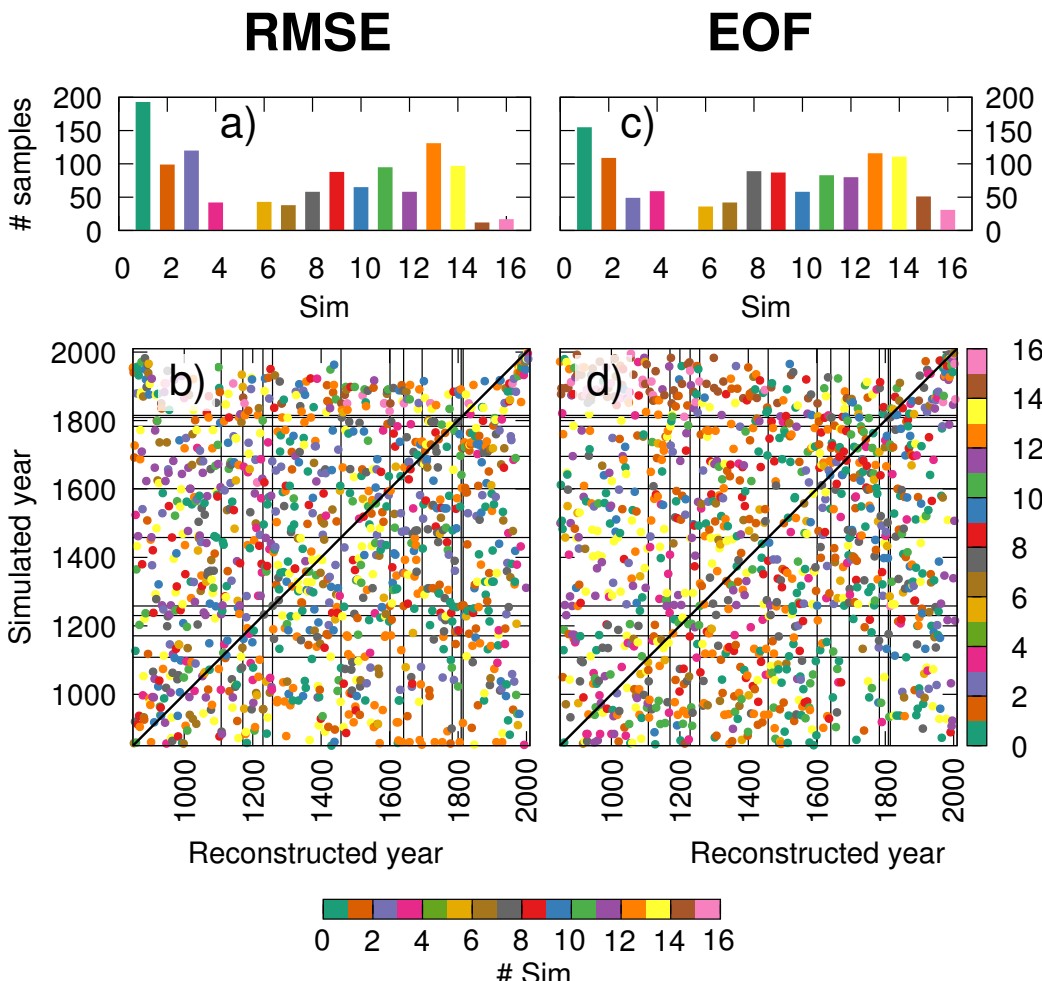

**Figure 4.** Selection of analogs used to carry out a perfect PPE. Bars in panels a and c indicate the number of times the analog has been taken from each of the 16 models. The points in panels b and d indicate the relationship between the reconstructed year (x-axis) and the model (colour) and simulated year (y-axis) used as analog for the reconstruction. Black horizontal and vertical lines show the timing of major volcanic eruptions according to Sigl et al. (2015). a and b correspond to the reconstruction based on RMSE and c and d based on Euclidian distance in the EOF space.

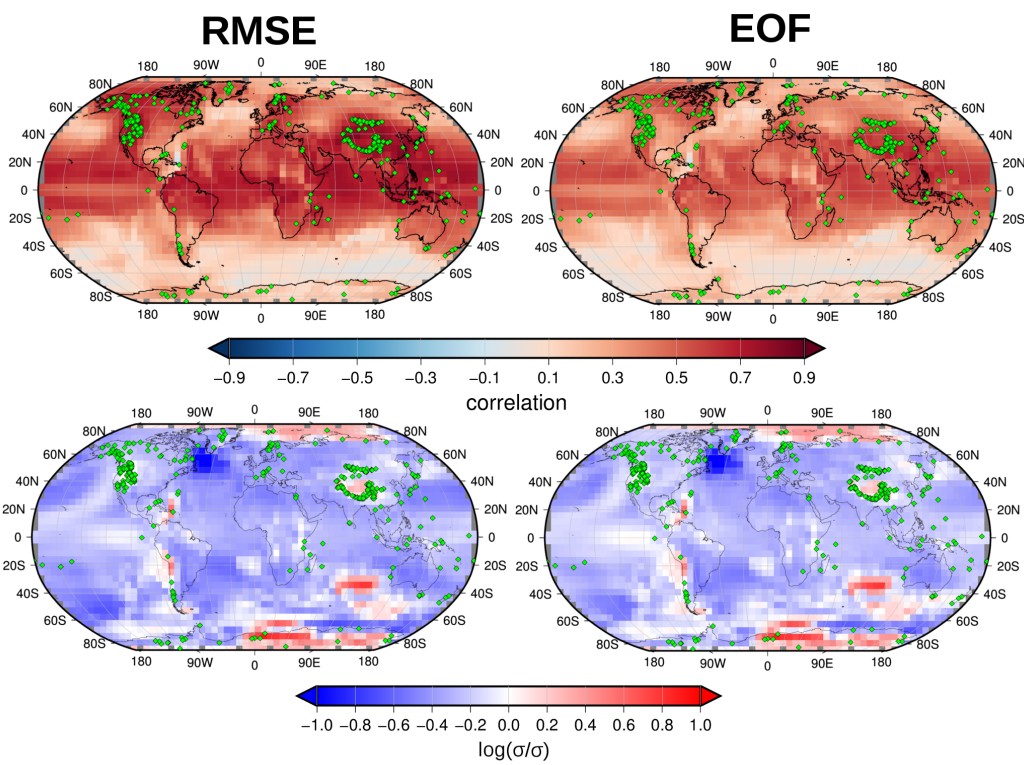

**Figure 5.** Similar to Figs. 2 and 3 but for realistic PPE. Top (bottom) row indicate the correlation (ratio of standard deviations) between the original simulation used as target and the reconstructions obtained selecting analogs from the PMIP3 pool.

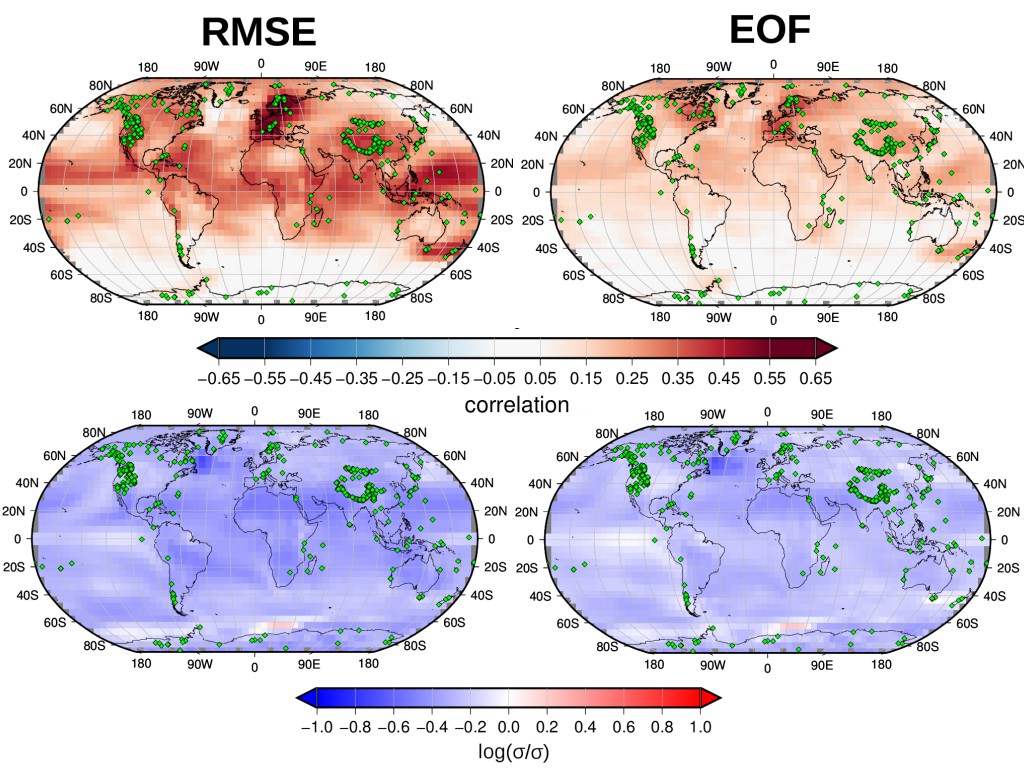

**Figure 6.** As Fig. 5 but for the hyper realistic PPE in which the correlations equal the values obtained during the proxies calibration, i.e. Fig. 1.

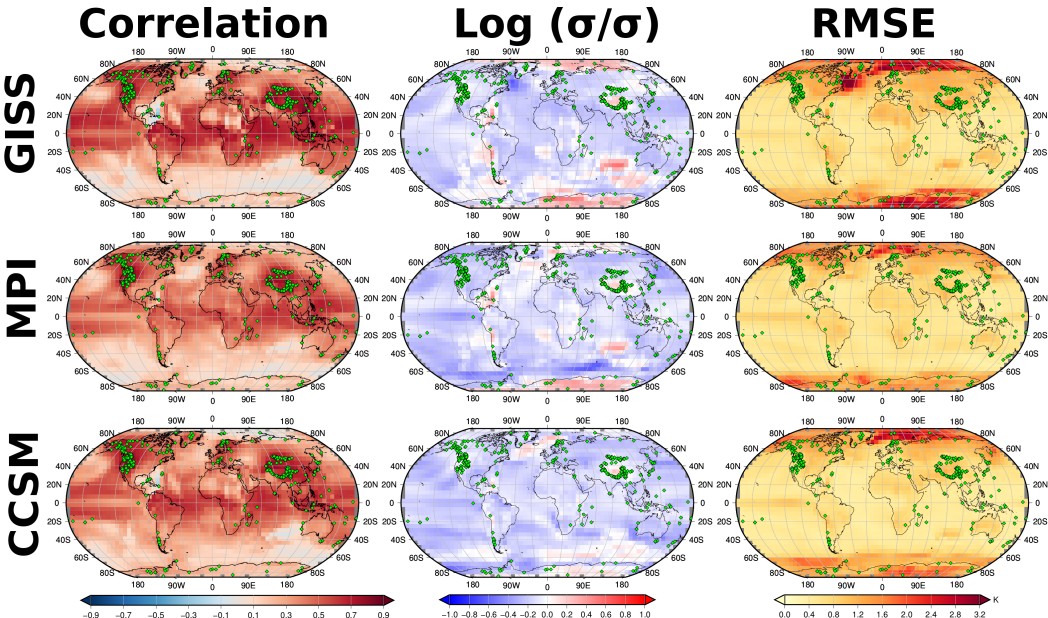

**Figure 7.** Correlation (left), logarithm of the ratio of the standard deviations (middle) and RMSE (right) between the target SAT and the pseudo-reconstructed SAT based in a PPE with additive white noise as in Section 4.2. All reconstructions use the same AM setup based on searching analogs that minimises RMSE and then average the 5 closest analogs. The only difference across rows is the model used as target for the PPE: GISS (top map, equivalent to Fig. 5) MPI-ESM-P (middle) and CESM4 (bottom).

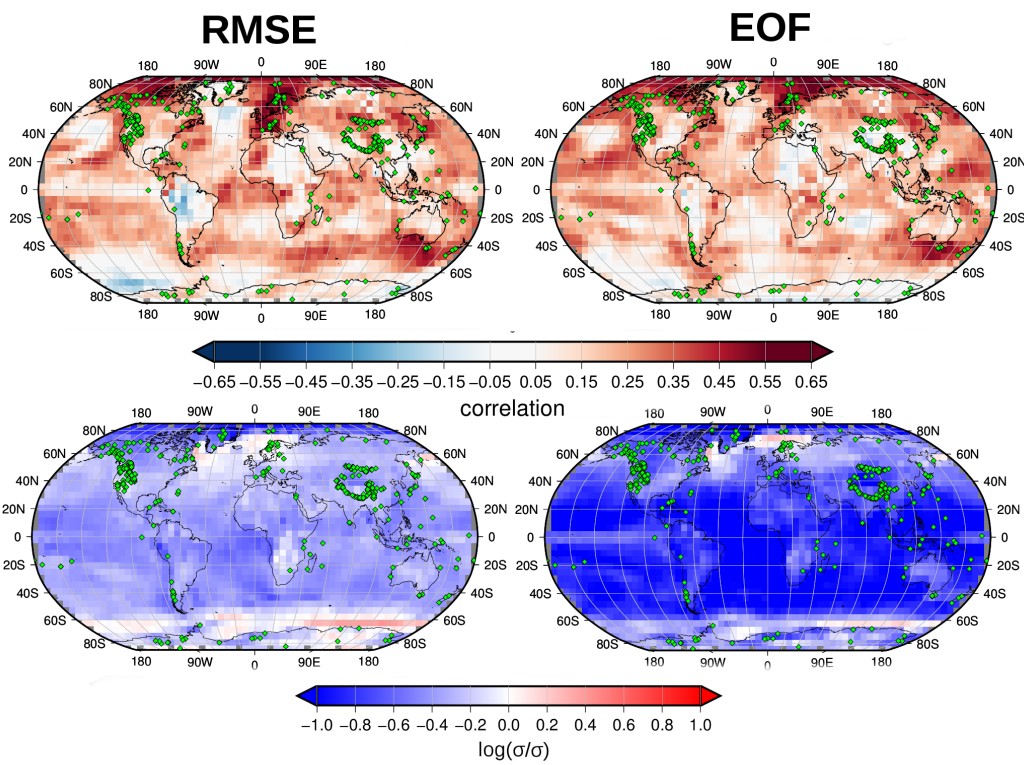

**Figure 8.** Similar to Figs. 5 and 6, but for a reconstruction of observations based on a calibration of proxies in the period 1911-1995. The correlation is calculated for the period 1850-2010.

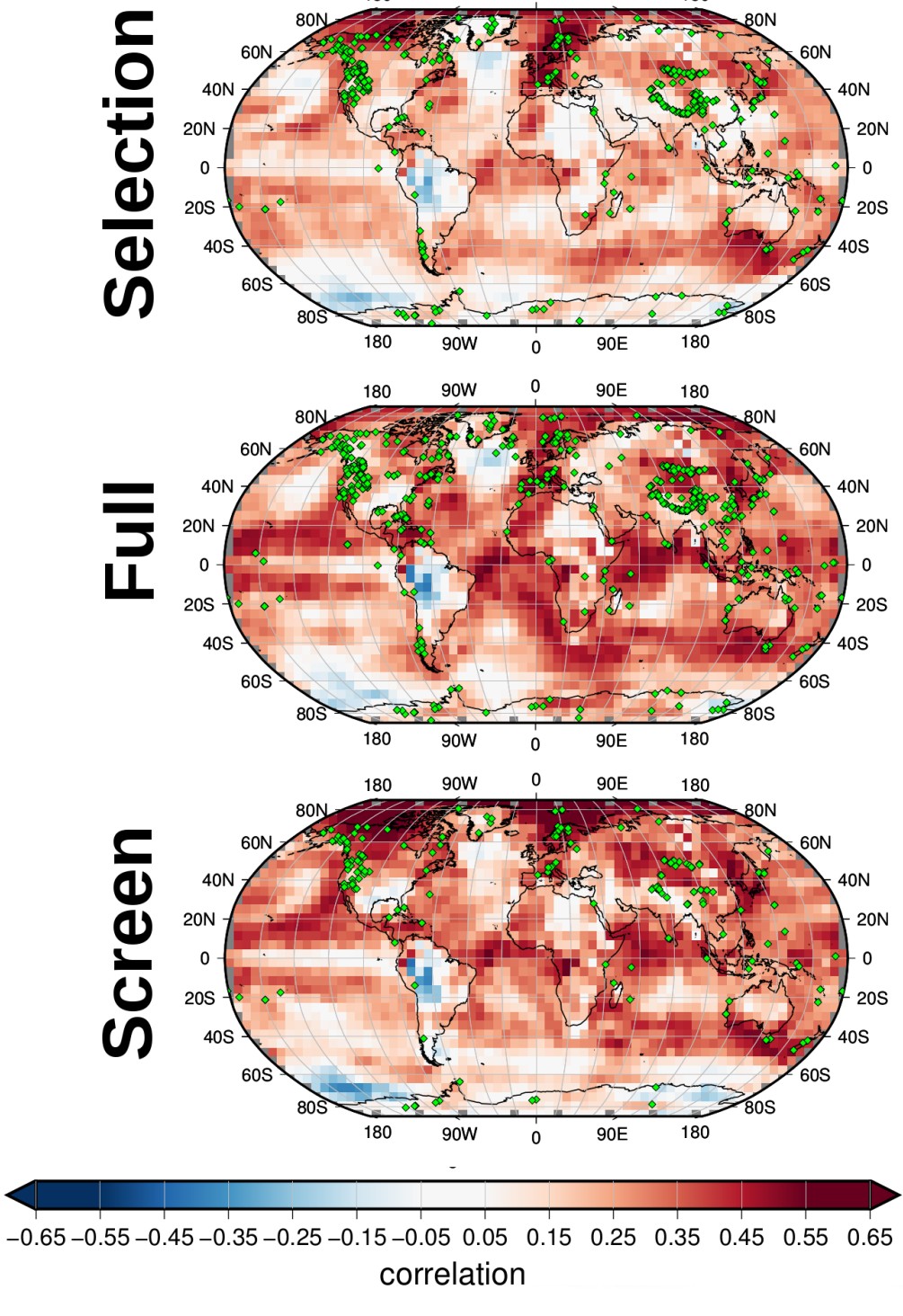

**Figure 9.** Correlation maps similar to Fig. 8 for the RMSE-AM variant of the AM method. The three maps depict the result obtained using each of the three variants of the PAGES2K network described in Section 2.2. In all cases the green symbols indicate the location of the proxies employed to reconstruct.

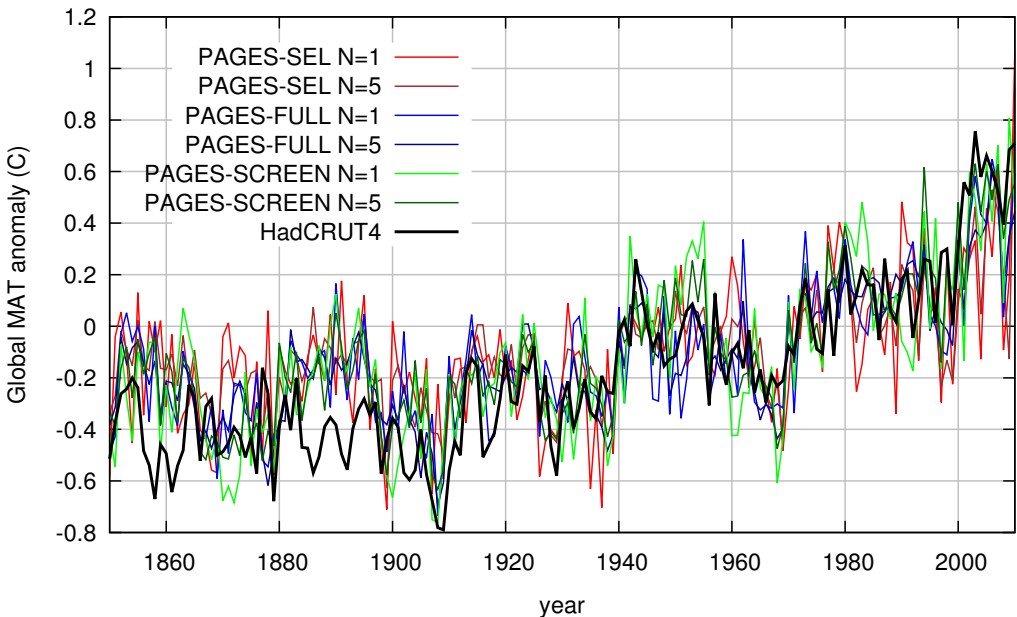

**Figure 10.** Time series of globally averaged SAT anomalies with respect to the period 1961-1990. The black bold line represents the infilled HadCRUT4 dataset, whereas colours indicate 6 reconstructions based on $N = 1, 5$ in Eq. 2 using the RMSE-AM version with the three variants of the PAGES2K network described in Section 2.2.

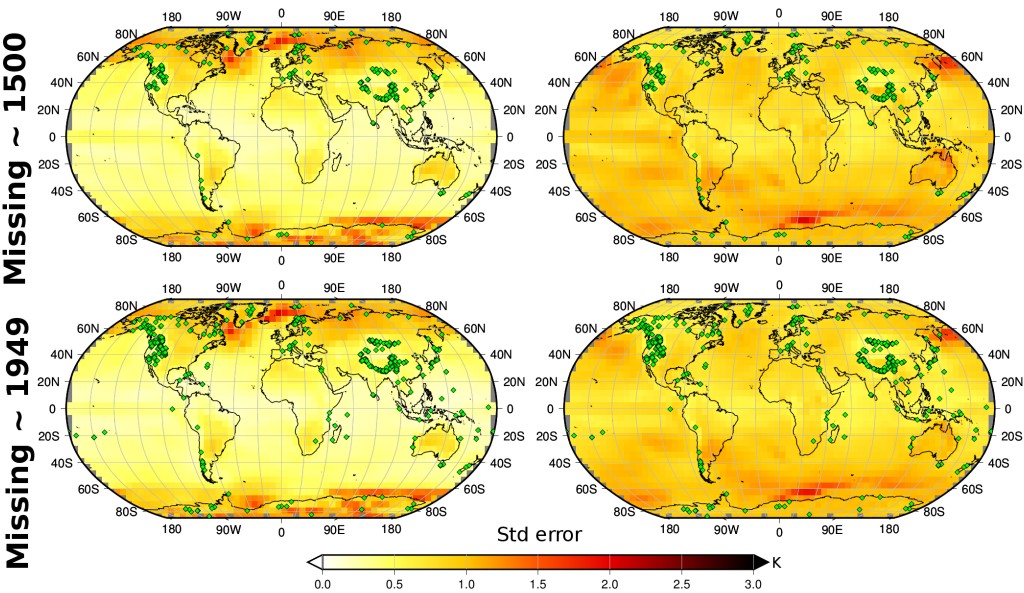

**Figure 11.** Left column: local standard deviation of the residuals (GISS r1i1p1 annual mean SAT minus pseudo-reconstructed SAT) over the period 1850-2005. Top: using a pseudo-proxy network with as many missing values as the PAGES-SEL network in 1500 (257 records). Bottom: using the maximum number of pseudo-proxy locations of the same network, which happens in 1949 (514 records). Right column: same as left column, but normalised by the standard deviation of the target. The precise location of the pseudo-proxies is indicated with green symbols.