# Peer review of "Pseudo-proxy tests of the analogue method to reconstruct spatially resolved global temperature during the Common Era"

_Climate of the Past, 2016_

## Short Comment (SC1) · 31 Oct 2016

This is an interesting paper. I don't intend to give a comprehensive review but would like to add a few comments.

The method of analogue selection seems very similar to what Goosse and co-workers have presented and described in a large number of papers (dating back at least to 2006) as a simplified particle filter, so I'm surprised that this does not come up for discussion anywhere in the current manuscript. Admittedly the method of Goosse et al does differ slightly in that they usually re-initialise the model from the selected best state in order to generate the prior for the subsequent year, but since there is very little predictive skill on the annual time scale, this difference in approach is relatively

unimportant (as the authors here also note). Our own work (Annan and Hargreaves Climate of the Past 2012) also supports this view. While Goosse et al usually use a sample size of around 100 ensemble members, these do all share an appropriate external forcing whereas a large majority of the CMIP/PMIP ensemble of opportunity used here will consist of simulated years when the forcing and its recent history is (relatively) far from the truth. This will not matter if the forcing is small enough to be unimportant, of course, which may be the case for much of the last millennium. Our own work used an ensemble of approximately 10,000 model years in some idealised testing and found it inadequate for reconstructing local temperatures but useful on the hemispheric scale. It is not clear from the presentation of the results to what extent this is also the case in this manuscript and I suspect that some more examination of the EOF analogue might be instructive in this respect, since the adequacy of ensemble size for such particle based methods depends strongly on the dimensionality of the problem.

Although correlation is widely used as an assessment of reconstruction performance in the paleoclimate community, it is not standard in data assimilation and reanalyses and has the potential to convey a somewhat rosy view (in my opinion) of the actual quality of the results. I suggest that the authors also calculate the skill in more conventional terms, e.g. reduction in RMS error compared to the prior (in this case being the simple mean of the ensemble of model runs), both locally and hemispherically if that is desired.

Finally, the authors only seem to use one model (GISS) as the source of truth. I wonder if this might have biased the results, e.g. if GISS is atypical in some respect. If not too expensive, I'd hope the authors could repeat the experiments (or at least a subset of them) using a much wider range of models as ground truth.

———————————————————

---

## Referee Comment (RC1) · E. Boucher (Referee) · 30 Nov 2016

Dear editor of Climate of the Past;

I have carefully read the manuscript entitled "Testing the analog method in reconstructing the global mean annual temperature during the Common Era" by Gomez-Navarro et al, submitted for publication in Climate of the Past.

The paper proposes to test the suitability of the Analog Method (AM) for producing climate field reconstructions (CFR) of mean annual temperatures across the globe. The manuscript does not intend to fully compare pros and cons of the AM-CFR method with those of other methods available for producing CFR (eg regression-based methods or

REGEM algorithm). Instead, the authors make use of pseudo-proxies experiments (PPE) at various degrees of signal degradation (with climate models being the original reference signal) to pinpoint and eventually attribute problems relating to the method itself and to the quality / diversity / scarcity of the proxies used to find temperature analogs.

The paper is fairly well written and logical. It builds a strong and convincing argument and clearly shows that the AM-CFR method is well suited to produce spatially complete climate CFR reconstructions at the global scale. The paper builds on a strong literature review where the various concepts associated with CFR reconstruction are explicitly and precisely detailed. The mathematical demonstrations associated with the CFR method are precise, although sometimes not always necessary (correlation eq and RMSE Eq), but still, they reflect the high level of technical know-how of the authors with respect to AM-CFR approaches. I am confident that the paper will become and important reference for the use of CFR for global scale reconstructions

I have a few comments listed below, I would invite the authors to reply to them in order to clarify some aspects of the manuscript, most particularly the discussion of concepts relating to the AM method. The comments are somehow minor, but I expect the authors to address them in order to improve the general quality of the manuscript.

Comments

1) Spectral signature of proxies vs PPE. One of the most important finding in this article is without a doubt the fact that the AM method performs better when PPE are used instead of real world proxies. Indeed, real-world proxies seem to be noisier than expected and for that reason the search for analogs tends to be less accurate (than that performed with PPE), resulting in estimations that are less well correlated with observed MAT. However, a major difference between PPE and real-world proxies is that real-world proxies commonly encompass only parts of the full spectrum of variability of the MAT. For example, tree rings, especially when severely standardized, relate to

high frequency variations in MAT and while a correlation exists with that climatic field, it reflects mostly the correlation of high-frequency periodicities. Contrastingly, low frequency proxies (such as pollen) may correlate with MAT, assuming that the correlation is a valid statistic in the presence of auto-correlated series. –and this is not a trivial assumption-. Still, if significant, the correlation would reflect only the low frequency component of the variability in MAT. On top of this, the signature of high-frequency signals probably reflects characteristic of local to regional climate dynamics while lower frequency variations common to proxies and MAT (whether they are internally or externally forced) tend to be visible at larger spatial scales. (and that would make it easier for the AM method)

On the contrary, PPE resulting from the degradation (addition of white noise) of original climate fields might better preserve the original spectrum of variability characteristic of temperatures series because the added noise tends to propagates evenly throughout the spectrum of variability of original climate fields. In short, I find the discussion (L. 505) around the fact that real-world proxies exhibit lower than expected correlations values with MAT rather incomplete. I invite the authors to fully explore other factors responsible for this drop in correlation and to include these in the discussion. I suggest, at first glance, to first ensure that PPE, real-world proxies and components. I fear that considering only the correlation (between proxies and MAT) when generating PPE might cast shadow on other sources of noise and variability inherent to the proxies and might be one of reasons why the AM performs less well with real-world proxies than with PPE

2) Estimation of analogs with low-resolution proxies. The PAGES-FULL dataset contains proxies covering the last 2k around the world. Most of these proxies are highly resolved tree ring series that are clustered in few sites such as the Himalayas, subarctic Canada, western Europe and the Andes. The rest of the proxies found in PAGES –FULL are rather poorly resolved proxies with uncertain dating (pollen and others), during the Common Era. Those proxies contain numerous missing values (actually probably more missing values than observed values during the CE). Early in the manuscript (L. 180) the authors state that they interpolate between observed / dated values to emulate an annual resolution. But I fear that once these series are interpolated, they remain low-resolution proxies in the sense that each year is strongly autocorrelated to the previous or following one and so on, while the high frequency component remains inexistent. When strongly autocorrelated proxies are used to calculate analogs, correlations between observations and predictions are often overestimated (see Guiot et al 2010). This is because the best analog is most often found just before or after the given year for which an analog is required. Consequently, this property does not guarantee that an independent analog could be found for a more distant period. Without a proper assessment of this problem in the paper, I fear that low frequency proxies could contribute to artificially "boost" the correlations between predicted and observed MAT. As a matter of fact, figure 8 shows that the addition of a few low frequency proxies from PAST-SEL to PAST-FULL (N passing from 514 to 641 proxies) actually seems to boost prediction accuracies significantly especially at places where very few proxies are available. What part of this increased prediction accuracy comes from the above-mentioned effects?

3) Not clear how the AM-CFR achieves extrapolation and how reliable it is. At many occasions, the authors claim that the AM method is able to "extrapolate" in order to produce spatially complete MAT fields. I am not sure how this word is used here. To extrapolate means making a prediction "above the limits" of a given calibration dataset. If this is done in the present paper, I request that the authors produce additional analysis to demonstrate how and how well this is done. However, as I understand it (and maybe I am wrong here), the AM cannot extrapolate over the range of observed variability. By definition, an analog is a year (or a pool of years) observed during the Common Era where proxies are similar to those observed in the past, similarity being of course measured by some distance metric. So, in other words, the analog must exist during the Common Era for it to be transferred to the past. Consequently, an analog cannot extrapolate over the bounds of the variability in MAT. On the contrary,

the search for analogs is constrained within the bounds of observed variability, resulting in a native incapacity to extrapolate. Therefore, I thus suggest to replace the word extrapolation by the word "prediction" which is more what MA actually does. Following that same idea, I wonder what would be the consequences of an inability of the AM method to extrapolate over the range of observed variability. Since the method aims at producing climate field reconstructions, would it be well suited to reconstruct periods (eg the MWP) that could have been warmer (even locally) than the Common Era? What about the LIA? Would the method be able to reconstruct periods that are much colder that any of the last decades ? So given this discussion, is the method doomed to be extremely conservative? I would suggest to test that, for example, by removing the anomalously warm last decades visible in figure 9, calibrating the AM say during the 1911-1980 period, and predicting MAT for the 1980-201X period. Since this latter period is censored off from the calibration period, is the AM method able to predict it or does it simply underestimates it? This could orient the discussion on the capacity of the AM-CFR method to extrapolate

4) Evaluation of uncertainty. The uncertainty of reconstructions produced by the AM-CFR method is never shown or discussed. Is this uncertainty larger where proxies are inexistent? How large are the 95% confidence intervals and how dependent are they on the number / density of proxies. Could the authors add uncertainty bands around reconstructions in Figure 9 and / or produce a map of the size of the 95% confidence interval of the reconstructions? That would help measure the robustness and reliability of the AM-CFR method

Specific comments Figure 1. I would like these points to show which proxies (tree rings, pollen, ice cores, lake sediments) are located where, perhaps on figure 1b? L70. Able to produce (no d) L104. Cannot result (remove s)
* * *

---

## Referee Comment (RC2) · Anonymous Referee #2 · 2 Dec 2016

This paper introduces an anolg method and uses it to reconstruct spatial fields of temperature covering the whole globe. The paper looks at several variants of the analog technique and evaluates which performs best. Although not the first time an anolog method has been used to reconstruct past climate this study is a useful addition to the field and I found that it is interesting, provides a good review of the field, is well written and is logical and clear. As such I would definitely like to see this paper published. I do however first have several comments which I hope will improve the paper – I've split the comments into major and minor, although not all the major comments are that major.

Major comments (in no particular order)

I find the title and to a lesser extent the abstract to be slightly misleading. The title

suggests that the paper focuses on reconstructing global mean temperature – however this is only briefly mentioned in the main paper (fig 9 and one paragraph at the end of section 5) with the focus instead on spatial reconstructions. While I have no problem with the emphasis of the paper, I do think that the title should be changed accordingly or alternatively more emphasis should be placed in the analysis of the global mean temperature to make for a more consistent paper.

Equally I find the use of the acronym "MAT" slightly confusing. It is introduced as "global near-surface mean annual temperature (MAT)" - l139, but is used frequently to refer to local temperatures and line 501 mentions the "annual series of MAT". I'm therefore unsure what the M in MAT actually refers to (is it a global mean or an annual mean?). Personally I would prefer the less ambiguous SAT (surface air temperature) to be used. But MAT would also be OK if properly defined and consistently used.

One interesting finding of the paper is that the simulations and reconstruction of the Arctic has reduced variance compared to the observations. This is however a region where there is no or little coverage in the HadCRUT4 dataset. How sensitive is the result therefore to the infilling technique used? Would the results be changed if a different infilling was used e.g. that of Cowtan and Way 2014?

Since you discuss the sensitivity to individual models I think it would be useful to mention explicitly which models correspond to which bar in fig. 4. I think I can more or less piece it together based on the text but would like this stated clearly either in the text, figure caption or a separate table. When doing this the GISS model used as the target could be number 16 (or 1) so that it can be left off of figure 4 to avoid confusion. I would also be tempted to group colours by model e.g. make all MPI-ESM models different shades of green – as I think this would improve clarity. I think figure 4 would also benefit from panels b and d being square with a straight line through x=y, to highlight the important point that the results are not scattered around this line, except for the end of the 20th century. Is this also the case for volcanic years?

The majority of the results presented are obtained using only one model (GISS r1i1p121) as the target. In the text you describe this model as unusual in having low variance. I think it would therefore strengthen the paper if analysis was also carried out using all the other models (or at least some) as targets. I don't envisage all the results actually being shown in the paper, since this would be far too many, but I do think a statement saying that the results are not sensitive to the model used as a target would be very useful (if this is indeed the case).

In the theory or discussion section no mention is made of how this method could be used to provide uncertainties to the reconstructions. Would it be possible to add some information regarding how the analyses presented here could be used to provide an uncertainty estimate on the reconstruction, for future use?

No mention is also made regarding the drop in coverage of the proxy network back through time. This surprised me as I thought this would be one of the key considerations when reconstructing the climate of the past millennium and beyond. Given that some of your PPE experiments have changing coverage through time could you show the performance of your method as a function of time. Equally I think it would be very informative to include a case in figure 8 (and section 6) with a sparse proxy coverage reflecting, for example, long proxies which cover the whole period 1000-2000 (PAGES-1000?), as this would better reflect the performance of the method during these earlier periods when less data is available.

In section 5 you comment that the difference between figure 7 and figure 5 & 6 suggests that the level of noise employed in the first PPE is an underestimation. Could it not also be due to errors in the model fields?

If one of the goals of this paper is to lay the ground work for a future analysis which reconstructs the global mean climate of the last 1000 or 2000 years then I think that a comparison of your global annual mean reconstruction (fig 9) with simpler commonly used reconstructions methods which only use the proxies scaled to the observations

to make a composited reconstruction, as well as a comparison to just the raw model results would be very useful. This would then allow the reader some sense of how much the method presented here adds to simply using the proxies or the models on their own. I would consider that at least some improvement over both of these would be the minimum requirement for applying this to the climate of the past to produce a global mean reconstruction, although I appreciate, as made clear in the paper, that this method does add much more valuable spatial and multi-variable information and is not focused on just producing a global mean reconstruction. If this is not a goal of the analysis than this should be clearly stated.

Minor comments:

M in eq 4 is not defined.

It would be good to state what value of L is used i.e. how many EOFs are retained.

L3 – explain acronym AM-CFR

L4 -> As a test bed

L6 –> simulations from PMIP3 are used

L10-> provided by the PMIP3 ensemble

L99-> with respect to

L395 -> especially

L556 – tree -> three?

Figs2 and others – what period is the correlation calculated over?

Reference Cowtan, K. and Way, R. G. (2014), Coverage bias in the HadCRUT4 temperature series and its impact on recent temperature trends. Q.J.R. Meteorol. Soc., 140: 1935–1944. doi:10.1002/qj.2297

---

## Editor Comment (EC1) · J. Guiot (Editor) · 13 Dec 2016

Dear authors,

We have now the comments of two reviewers. Both are positive but they request corrections before the paper can be published. I ask you to post answers to these comments and to prepare a revised manuscript (in track change mode) that you will include with the answers.

Best regards

Joel Guiot, editor

---

## Author Comment (AC1) · 13 Jan 2017

*This is an interesting paper. I don't intend to give a comprehensive review but would like to add a few comments.*

The authors thank the time and interest shown in reading our manuscript and making constructive suggestions.

*The method of analogue selection seems very similar to what Goosse and co-workershave presented and described in a large number of papers (dating back at least to2006) as a simplified particle filter, so I'm surprised that this does not come up for discussion anywhere in the current manuscript. Admittedly the method of Goosse*

[Figure]

*et al does differ slightly in that they usually re-initialise the model from the selected best state in order to generate the prior for the subsequent year, but since there is very little predictive skill on the annual time scale, this difference in approach is relatively unimportant (as the authors here also note). Our own work (Annan and Hargreaves Climate of the Past 2012) also supports this view. While Goosse et al usually use a sample size of around 100 ensemble members, these do all share an appropriate external forcing whereas a large majority of the CMIP/PMIP ensemble of opportunity used here will consist of simulated years when the forcing and its recent history is (relatively) far from the truth. This will not matter if the forcing is small enough to be unimportant, of course, which may be the case for much of the last millennium. Our own work used an ensemble of approximately 10,000 model years in some idealised testing and found it inadequate for reconstructing local temperatures but useful on the hemispheric scale. It is not clear from the presentation of the results to what extent this is also the case in this manuscript and I suspect that some more examination of the EOF analogue might be instructive in this respect, since the adequacy of ensemble size for such particle based methods depends strongly on the dimensionality of the problem.*

Certainly we overlooked these references. We will re-formulate part of the introduction and the discussions of the results to improve these aspects of the manuscript. As pointed out by J. Annan, there are some differences between Goose's method and ours. One important difference is that Goose et al use in their work a climate model of intermediate complexity, with a fairly coarse spatial resolution - this is the price paid to be able to run a relatively large ensemble of simulations. The price we pay is that, as also pointed out by J.Annan, our analogues are not tied to the truth by a common external forcing. The analogue reconstruction is just driven by the available proxy data. Assessing to what extent this is permissible and yields reasonable results is exactly the aim of this study. The rationale behind it is that the patterns of variations are sufficiently constrained by the available proxies, and are not so much dependent on the particular forcing that dominates in a given year, either solar variations, greenhouse gas, etc. This might not always hold, but the study is essentially a test of this assumption.

*Although correlation is widely used as an assessment of reconstruction performance in the paleoclimate community, it is not standard in data assimilation and reanalyses and has the potential to convey a somewhat rosy view (in my opinion) of the actual quality of the results. I suggest that the authors also calculate the skill in more conventional terms, e.g. reduction in RMS error compared to the prior (in this case being the simple mean of the ensemble of model runs), both locally and hemispherically if that is desired. Finally, the authors only seem to use one model (GISS) as the source of truth. I wonder if this might have biased the results, e.g. if GISS is atypical in some respect. If not too expensive, I'd hope the authors could repeat the experiments (or at least a subset of them) using a much wider range of models as ground truth.*

We used additional performance metrics, including RMSE, but decided focus on just one for the sake of brevity and clarity in the manuscript. We chose correlation for being a rather intuitive metric but we will include other measures of skill in the revised version.

Regarding the choice of GISS as target, we will evaluate the impact of using alternative targets and include the discussion of this result in the manuscript. We believe however that the results will be consistent, as there exists also consistency we show between the results in PPE and in the calibration period with actual data.

---

## Author Comment (AC2) · 13 Jan 2017

*Dear editor of Climate of the Past; I have carefully read the manuscript entitled "Testing the analog method in reconstruct- ing the global mean annual temperature during the Common Era" by Gomez-Navarro et al, submitted for publication in Climate of the Past.*

*The paper proposes to test the suitability of the Analog Method (AM) for producing climate field reconstructions (CFR) of mean annual temperatures across the globe. The manuscript does not intend to fully compare pros and cons of the AM-CFR method with those of other methods available for producing CFR (eg regression-based methods orREGEM algorithm). Instead, the authors make use of pseudo-proxies experiments (PPE) at various degrees of signal degradation (with climate models being the original*

[Figure]

*reference signal) to pinpoint and eventually attribute problems relating to the method itself and to the quality / diversity / scarcity of the proxies used to find temperature analogs.*

*The paper is fairly well written and logical. It builds a strong and convincing argument and clearly shows that the AM-CFR method is well suited to produce spatially complete climate CFR reconstructions at the global scale. The paper builds on a strong literature review where the various concepts associated with CFR reconstruction are explicitly and precisely detailed. The mathematical demonstrations associated with the CFR method are precise, although sometimes not always necessary (correlation eq and RMSE Eq), but still, they reflect the high level of technical know-how of the authors with respect to AM-CFR approaches. I am confident that the paper will become and important reference for the use of CFR for global scale reconstructions*

*I have a few comments listed below, I would invite the authors to reply to them in order to clarify some aspects of the manuscript, most particularly the discussion of concepts relating to the AM method. The comments are somehow minor, but I expect the authors to address them in order to improve the general quality of the manuscript.*

The authors are grateful for the time devoted to the review of the article and the positive view of it

*1) Spectral signature of proxies vs PPE. One of the most important finding in this article is without a doubt the fact that the AM method performs better when PPE are used instead of real world proxies. Indeed, real-world proxies seem to be noisier than expected and for that reason the search for analogs tends to be less accurate (than that performed with PPE), resulting in estimations that are less well correlated with observed MAT. However, a major difference between PPE and real-world proxies is that real-world proxies commonly encompass only parts of the full spectrum of variability of the MAT. For example, tree rings, especially when severely standardized, relate to high frequency variations in MAT and while a correlation exists with that climatic field,*

*it reflects mostly the correlation of high-frequency periodicities. Contrastingly, low frequency proxies (such as pollen) may correlate with MAT, assuming that the correlation is a valid statistic in the presence of auto-correlated series. –and this is not a trivial assumption-. Still, if significant, the correlation would reflect only the low frequency component of the variability in MAT. On top of this, the signature of high-frequency signals probably reflects characteristic of local to regional climate dynamics while lower frequency variations common to proxies and MAT (whether they are internally or externally forced) tend to be visible at larger spatial scales. (and that would make it easier for the AM method).*

*On the contrary, PPE resulting from the degradation (addition of white noise) of original climate fields might better preserve the original spectrum of variability characteristic of temperatures series because the added noise tends to propagates evenly throughout the spectrum of variability of original climate fields. In short, I find the discussion (L. 505) around the fact that real-world proxies exhibit lower than expected correlations values with MAT rather incomplete. I invite the authors to fully explore other factors responsible for this drop in correlation and to include these in the discussion. I suggest, at first glance, to first ensure that PPE, real-world proxies and components. I fear that considering only the correlation (between proxies and MAT) when generating PPE might cast shadow on other sources of noise and variability inherent to the proxies and might be one of reasons why the AM performs less well with real-world proxies than with PPE*

This is an excellent suggestion, although some members of the dendroclimatological community may disagree on the extent of variance loss at low-frequencies by tree-ring proxies. Relatively modern standardization methods, like RCS, claim a much better variance preservation properties than 'classical' standardization methods. However, the question posed by the reviewer is indeed relevant, since it is not always clear what the spectral properties of proxies, and of their non-climate noise, are. According to previous publications with very a similar methods, but for reconstructing regional pre-

cipitation over Europe, addressed this issue (Gómez-Navarro et al., 2014; cited in the manuscript). The authors found that the difference between white and red noise is in the design of PPE is rather minor. Still we will explore and report the results in the revised version including additional PPE with non-white noise, trying to mimic the behavior of proxy records. Note however that this will always be incomplete, as it is difficult to cover all possible types of proxies available and the wealth of varieties of possible non-climatic noise.

*2) Estimation of analogs with low-resolution proxies. The PAGES-FULL dataset contains proxies covering the last 2k around the world. Most of these proxies are highly resolved tree ring series that are clustered in few sites such as the Himalayas, subarctic Canada, western Europe and the Andes. The rest of the proxies found in PAGES–FULL are rather poorly resolved proxies with uncertain dating (pollen and others), during the Common Era. Those proxies contain numerous missing values (actually probably more missing values than observed values during the CE). Early in the manuscript (L. 180) the authors state that they interpolate between observed / dated values to emulate an annual resolution. But I fear that once these series are interpolated, they remain low-resolution proxies in the sense that each year is strongly autocorrelated to the previous or following one and so on, while the high frequency component remains inexistent. When strongly autocorrelated proxies are used to calculate analogs, correlations between observations and predictions are often overestimated (see Guiot et al 2010). This is because the best analog is most often found just before or after the given year for which an analog is required. Consequently, this property does not guarantee that an independent analog could be found for a more distant period. Without a proper assessment of this problem in the paper, I fear that low frequency proxies could contribute to artificially "boost" the correlations between predicted and observed MAT. As a matter of fact, figure 8 shows that the addition of a few low frequency proxies from PAST-SEL to PAST-FULL (N passing from 514 to 641 proxies) actually seems to boost prediction accuracies significantly especially at places where very few proxies are available. What part of this increased prediction accuracy comes from the above-mentioned effects?*

We believe that the point raised by the reviewer is not really an issue in this study. It is important to recall that the low frequency in the proxies is in principle unrelated to its counterpart in the pool (actually the pool has no temporal order by definition, so the concept of autocorrelation does not apply). Further, the search of analogs is carried out independently in each time step, disregarding previous time steps. Thus, the fact that some proxies might have artificially high levels of autocorrelation does not boost artificially the correlation between the target and the analog. This is especially true in the case of PPE, where such artificial autocorrelations do not exist and no interpolation is carried out to remove missing values.

*3) Not clear how the AM-CFR achieves extrapolation and how reliable it is. At many occasions, the authors claim that the AM method is able to "extrapolate" in order to produce spatially complete MAT fields. I am not sure how this word is used here. To extrapolate means making a prediction "above the limits" of a given calibration dataset. If this is done in the present paper, I request that the authors produce additional analysis to demonstrate how and how well this is done. However, as I understand it (and maybe I am wrong here), the AM cannot extrapolate over the range of observed variability. By definition, an analog is a year (or a pool of years) observed during the Common Era where proxies are similar to those observed in the past, similarity being of course measured by some distance metric. So, in other words, the analog must exist during the Common Era for it to be transferred to the past. Consequently, an analog cannot extrapolate over the bounds of the variability in MAT. On the contrary, the search for analogs is constrained within the bounds of observed variability, resulting in a native incapacity to extrapolate. Therefore, I thus suggest to replace the word extrapolation by the word "prediction" which is more what MA actually does. Following that same idea, I wonder what would be the consequences of an inability of the AM method to extrapolate over the range of observed variability. Since the method aims at producing climate field reconstructions, would it be well suited to reconstruct periods (eg the MWP) that could have been warmer (even locally) than the Common Era? What about the LIA? Would the method be able to reconstruct periods that are much*

*colder that any of the last decades? So given this discussion, is the method doomed to be extremely conservative? I would suggest to test that, for example, by removing the anomalously warm last decades visible in figure 9, calibrating the AM say during the 1911-1980 period, and predicting MAT for the 1980-201X period. Since this latter period is censored off from the calibration period, is the AM method able to predict it or does it simply underestimates it? This could orient the discussion on the capacity of the AM-CFR method to extrapolate*

We believe this issue is rather a misunderstanding, probably caused by not using the term "extrapolation" carefully enough. We do not mean extrapolation in a temporal sense, and certainly the analogs are just pool members (or plain averages of them). Therefore it is clear that the output of the reconstruction is constrained by the variability in the pool, so in this sense the method does not extrapolate anything outside the pool. Still, is worth to note that due to the multiple models merged and the length of each run, the available variability within the pool is larger than the variability observed in the calibration period.

In any case, what we mean when we write "extrapolation" in this context is "spatial extrapolation". It is an expression we use to refer to the general goal aimed by CFR techniques: filling spatial gaps between local reconstructions. Therefore we will re-write the parts where this expression is used to minimize chances of mislead the reader.

*4) Evaluation of uncertainty. The uncertainty of reconstructions produced by the AM-CFR method is never shown or discussed. Is this uncertainty larger where proxies are inexistent? How large are the 95% confidence intervals and how dependent are they on the number / density of proxies. Could the authors add uncertainty bands around reconstructions in Figure 9 and / or produce a map of the size of the 95% confidence interval of the reconstructions? That would help measure the robustness and reliability of the AM-CFR method*

This is an important caveat of the first version of the manuscript that has also been

pointed out by other reviewers. However, this question is actually much deeper than it may seem at first sight, since there is no theory -to our knowledge- to estimate estimation uncertainty in the AM as in other established statistical methods like linear regression.

There may be different sources of uncertainty, and the reviewer seems to be referring only to the uncertainty due to the limited number of proxy records. Other sources of uncertainty will be related to the finite spatial correlation of the temperature field (since the AM method is used to produce a full temperature field based on individual pseudo-proxy records. When using observations as analogue pool, thus source of uncertainty will be smaller than when using model output as analogue pool, since a climate model output will never perfectly represent 'reality'. Other sources of uncertainty are related to the similarity between the target pattern and the selected analogue: when target and analogue are very dissimilar because the analogue pool is too small, the uncertainty should be larger than when the target and the analogue are indeed similar.

To some extent, the sources of uncertainty not related to the proxy network are indirectly estimated with the PPE, comparing the range of reconstructions with the actual 'true' temperature, but of course this is valid only in the setting used here: the GISS model as target and other models as analogue pool. In an extreme case, where the proxy records cover the whole world, the uncertain will be mostly caused by the similarity between the spatial correlation of the 'true' field and the spatial correlation of the fields in the pool of analogues.

However we agree that an uncertainty estimate for our results should be provided. Developing a method that addresses all the above mentioned sources of uncertainty is beyond the scope of this study. Therefore we will produce an estimation of the uncertainty that is at least comparable to those used in linear regression-based CFR methods, following standard procedures that make use of the calibration residuals. We will visualize temporal and spatial changes in these uncertainties, as suggested.

*Specific comments Figure 1. I would like these points to show which proxies (tree rings, pollen, ice cores, lake sediments) are located where, perhaps on figure 1b? L70. Able to produce (no d) L104. Cannot result (remove s)*

We will consider these comments in the reviewed manuscript.

---

## Author Comment (AC3) · 13 Jan 2017

*This paper introduces analog method and uses it to reconstruct spatial fields of temperature covering the whole globe. The paper looks at several variants of the analog technique and evaluates which performs best. Although not the first time analog method has been used to reconstruct past climate this study is a useful addition to the field and I found that it is interesting, provides a good review of the field, is well written and is logical and clear. As such I would definitely like to see this paper published. I do however first have several comments which I hope will improve the paper – I've split the comments into major and minor, although not all the major comments are that major.*

Thank you very much for the time devoted to carefully read the manuscript and the

positive view expressed about it. We will try to address all major concerns pointed out by the reviewer.

*Major comments (in no particular order).*

*I find the title and to a lesser extent the abstract to be slightly misleading. The title suggests that the paper focuses on reconstructing global mean temperature – however this is only briefly mentioned in the main paper (fig 9 and one paragraph at the end of section 5) with the focus instead on spatial reconstructions. While I have no problem with the emphasis of the paper, I do think that the title should be changed accordingly or alternatively more emphasis should be placed in the analysis of the global mean temperature to make for a more consistent paper.*

We will re-think the title according to this suggestions and the changes carried out in the reviewed manuscript.

*Equally I find the use of the acronym "MAT" slightly confusing. It is introduced as "global near-surface mean annual temperature (MAT)" - l139, but is used frequently to refer to local temperatures and line 501 mentions the "annual series of MAT". I'm therefore unsure what the M in MAT actually refers to (is it a global mean or an annual mean?). Personally I would prefer the less ambiguous SAT (surface air temperature) to be used. But MAT would also be OK if properly defined and consistently used.*

We chose that acronym following the naming convention used some references we use. We will carefully reconsider what naming convention leads to the more natural reading for the expected target audience of the paper.

*One interesting finding of the paper is that the simulations and reconstruction of the Arctic has reduced variance compared to the observations. This is however a region where there is no or little coverage in the HadCRUT4 dataset. How sensitive is the result therefore to the infilling technique used? Would the results be changed if a different infilling was used e.g. that of Cowtan and Way 2014?*

[Figure]

Using a target infilled differently would have a minor role in the output of the reconstruction, as the analogs themselves are independent on this. The only effect would come from their choice, which can be affected by having obtained slightly different calibrations in the arctic area. But we believe this is a second order effect that is anyway masked by the weight of the rest of proxies worldwide.

However, what is more important is the fact that using a different target would have an important and direct effect on the metrics we obtain from the evaluation of the reconstruction. In particular, using the Cowtan and Way infilling as target as suggested by the reviewer will certainly reduce the underestimation of variance we report here. Therefore this is an important point we will develop in the discussion.

*Since you discuss the sensitivity to individual models I think it would be useful to mention explicitly which models correspond to which bar in fig. 4. I think I can more or less piece it together based on the text but would like this stated clearly either in the text, figure caption or a separate table. When doing this the GISS model used as the target could be number 16 (or 1) so that it can be left off of figure 4 to avoid confusion. I would also be tempted to group colours by model e.g. make all MPI-ESM models different shades of green – as I think this would improve clarity. I think figure 4 would also benefit from panels b and d being square with a straight line through x=y, to highlight the important point that the results are not scattered around this line, except for the end of the 20th century. Is this also the case for volcanic years?*

In the first version of the manuscript we decided not to label models on purpose, since the nature of the method consist of considering all models equally as members of a pool with all models contributing equally without any consideration of their intrinsic skill, and where the method selects years blindly without other consideration than the chosen metric. Therefore we are not sure if labeling models is sensible in this context. Still, we will carefully consider these suggestions and decide how to proceed giving our reasons in the final version of the manuscript.

*The majority of the results presented are obtained using only one model (GISS r1i1p121) as the target. In the text you describe this model as unusual in having low variance. I think it would therefore strengthen the paper if analysis was also carried out using all the other models (or at least some) as targets. I don't envisage all the results actually being shown in the paper, since this would be far too many, but I do think a statement saying that the results are not sensitive to the model used as a target would be very useful (if this is indeed the case).*

This is a comment shared by the other reviewer, so we will indeed carry out this test, although we will surely not include the figures to keep the paper as concise as possible.

*In the theory or discussion section no mention is made of how this method could be used to provide uncertainties to the reconstructions. Would it be possible to add some information regarding how the analyses presented here could be used to provide an uncertainty estimate on the reconstruction, for future use?*

We will enlarge the paper to provide a method to estimate uncertainties (see also our response to the above comment of E. Boucher regarding uncertainties). We envisage a method similar to those proposed in regression-based methods.

*No mention is also made regarding the drop in coverage of the proxy network back through time. This surprised me as I thought this would be one of the key considerations when reconstructing the climate of the past millennium and beyond. Given that some of your PPE experiments have changing coverage through time could you show the performance of your method as a function of time. Equally I think it would be very informative to include a case in figure 8 (and section 6) with a sparse proxy coverage reflecting, for example, long proxies which cover the whole period 1000-2000 (PAGES-1000?), as this would better reflect the performance of the method during these earlier periods when less data is available.*

We believe these are interesting suggestion. We will consider additional tests to explore the sensitivity of the performance to the variable number of proxies.

*In section 5 you comment that the difference between figure 7 and figure 5 6 suggests that the level of noise employed in the first PPE is an underestimation. Could it not also be due to errors in the model fields?*

This could be an alternative explanation we will consider in the discussion.

*If one of the goals of this paper is to lay the ground work for a future analysis which reconstructs the global mean climate of the last 1000 or 2000 years then I think that a comparison of your global annual mean reconstruction (fig 9) with simpler commonly used reconstructions methods which only use the proxies scaled to the observationsto make a composited reconstruction, as well as a comparison to just the raw model results would be very useful. This would then allow the reader some sense of how much the method presented here adds to simply using the proxies or the models on their own. I would consider that at least some improvement over both of these would be the minimum requirement for applying this to the climate of the past to produce a global mean reconstruction, although I appreciate, as made clear in the paper, that this method does add much more valuable spatial and multi-variable information and is not focused on just producing a global mean reconstruction. If this is not a goal of the analysis than this should be clearly stated.*

We wanted to keep the paper as short as possible by 1) focusing on the performance of the method to reproduce the spatial patterns, rather than the global average and 2) not including a full range of alternative methodologies, as this work is indeed under preparation to be submitted as a separate publication. However we agree with the reviewer that such a comparison with CFR would not be very complex and could enrich the paper. Therefore we will include a comparison with a global mean reconstruction carried out with Composite plus Scaling.

*Minor comments:*

- *M in eq 4 is not defined.*

- *It would be good to state what value of L is used i.e. how many EOFs are retained.*

- *L3 – explain acronym AM-CFR*

- *L4 -> As a test bed*

- *L6 –> simulations from PMIP3 are used*

- *L10-> provided by the PMIP3 ensemble*

- *L99-> with respect to*

- *L395 -> especially*

- *L556 – tree -> three?*

- *Figs2 and others – what period is the correlation calculated over?*

We will carefully review these comments in the final version of the manuscript.

---

## Author Comment (AC4) · 13 Jan 2017

Thank you very much for the encouragement to carry out changes and submit a reviewed manuscript. We will have the reviewer's suggestions carefully into account in the new version.

---

## Author Response (AR2)

**Point by point authors response to reviewers**

J. J. Gómez-Navarro on behalf of all co-authors

March 6, 2017

**1 Comment by James Annan**

We thank James Annan for his interest in the manuscript and his helpful suggestions to expand/improve it. In the interactive open phase we had indicated how we planned to addressed these suggestions and basically we have followed this initial response in the revised version.

**1.1 Comparison with the particle filter method of Goose et al.**

We have re-formulated part of the introduction and the discussions of the results to improve these aspects of the manuscript. As pointed out by J. Annan, there are some differences between Goose's method and ours. One important difference is that Goose et al. use in their work a climate model of intermediate complexity, with a fairly coarse spatial resolution - this is the price paid to be able to run a relatively large ensemble of simulations. The price we pay is that, as also pointed out by J. Annan, our analogues are not tied to the truth by a common external forcing. The analogue reconstruction is just driven by the available proxy data and not by the forcing. The proxy data contain in part indirect information about the forcing, but at inter-annual timescales the variability of the proxies and also of the models is mostly driven by internal processes (with the exceptions of years with large volcanic eruptions). There are two underlying assumptions of the analogue method. One is that that the external forcing does not critically change the spatial patterns of temperature anomalies in the past and that externally forced patterns will have matching patterns that are internally generated. Alternatively, the analogue methods assumes that if the externally forced patterns are critically different from the patterns generated by internal variations, the analogue method will be able to identify also externally forced patterns in the pool of potential analogues. Assessing to what extent this is permissible and yields reasonable results, on average, is exactly the aim of this study. We have expanded the introduction and the discussions to highlight these points, which indeed expand the questions that can be addressed in further work, for instance, to what extent patterns forced by volcanism can be also found in control simulations, for instance.

**1.2 Using other metrics than correlation**

In the course of this investigation, we employed several metrics, and the results turn out to be consistent regardless of the metric employed. Further very little new insight is provided if other variables are considered in the discussion. This is consistent with previous experience of the authors in similar analyses, as shown in Gómez-Navarro et al. (2014). Therefore we have decided to limit the discussion to correlation to simplify the exposition of the results. Still, we have included a figure with RMSE in the new Figure 7 that illustrates this point.

**1.3 Using other models as 'truth' in addition to the GISS simulation**

This is a suggestions also raised by reviewer #2. We have performed and included in the manuscript additional tests with two additional models used as target truth with very similar results. We

therefore conclude that the choice of 'truth' within the CMIP5 ensemble is not critical to the overall conclusions we draw.

**2 Reviewer 1, Etienne Boucher**

We thank Etienne Boucher for his careful reading of the manuscript and his constructive suggestions to improve it. We have considered all of them in our revisions of the manuscript. All suggestions have lead to an expansion of the results and the discussion, with the exception of his comment #3, which is due to a misunderstanding of the text. Here, we have reworded the critical expression on 'extrapolation' to avoid any misunderstanding by the reader.

**2.1 Spectral signature of proxies vs PPE**

The reviewer suggests to also test the Analog method with red-noise pseudo-proxies. In our short comment in the interactive discussion we had stated that:

> *"This is an excellent suggestion, although some members of the dendroclimatological community may disagree on the extent of variance loss at low-frequencies by tree-ring proxies. Relatively modern standardization methods, like RCS, claim a much better variance preservation properties than 'classical' standardization methods. However, the question posed by the reviewer is indeed relevant, since it is not always clear what the spectral properties of proxies, and of their non-climate noise, are. According to previous publications with very a similar methods, but for reconstructing regional precipitation over Europe, addressed this issue (Gómez-Navarro et al., 2014; cited in the manuscript). The authors found that the difference between white and red noise is in the design of PPE is rather minor. Still we will explore and report the results in the revised version including additional PPE with non-white noise, trying to mimic the behavior of proxy records. Note however that this will always be incomplete, as it is difficult to cover all possible types of proxies available and the wealth of varieties of possible non-climatic noise."*

We have now performed those experiments with red-noise pseudo-proxies using a plausible de-correlation time of 5 years and a local correlation with the grid-cell temperature of 0.5. These settings should give a flavour of the impact of red-noise on the final pseudo-reconstructions, although we acknowledge that the real value of the decorrelation of the noise in real proxies is difficult to ascertain. Tree-ring proxies seem to indicate this order of magnitude for the noise de-correlation time, although there are some on-going discussions as to whether this decorrelation time may be at least in part induced by the decorrelation of the temperature itself. In any case, our PPE experiments confirm our previous experience that the inclusion of red-noise pseudo-proxies instead of white-noise pseudo-proxies is indeed very minor. We include some figures of those results here in Figure 1, but we feel that they look so similar to the results obtained with white noise pseudo-proxies that it suffice to describe them in writing in the manuscript, without including additional figures in the manuscript.

**2.2 Estimation of analogs with low-resolution proxies**

The reviewer suggests to change the network of pseudo-proxies with two goals: i) to take into account the nature of the real low-resolution proxies which are then interpolated in time. This could induce an artificial autocorrelation in the proxies and affect the selection of analogs. ii) To represent more realistically that the network of proxies diminishes back in time.

In our response in the interactive discussion we had indicated that:

> *"We believe that the point raised by the reviewer is not really an issue in this study. It is important to recall that the low frequency in the proxies is in principle unrelated to*

[Figure]

Figure 1: Comparative of the performance of the AM using PPE with white (left) versus red (right) noise. The figure shows the correlation (top) as well as RMSE (bottom) between the target and the pseudo-reconstruction when the same amount of noise is introduced to contaminate the proxies, i.e. it reduces the correlation to 0.5 as in the experiments R05 PPE described in Section 4.2 in the manuscript.

> *its counterpart in the pool (actually the pool has no temporal order by definition, so the concept of autocorrelation does not apply). Further, the search of analogs is carried out independently in each time step, disregarding previous time steps. Thus, the fact that some proxies might have artificially high levels of autocorrelation does not boost artificially the correlation between the target and the analog. This is especially true in the case of PPE, where such artificial autocorrelations do not exist and no interpolation is carried out to remove missing values."*

Indeed, unless we misunderstood this comment, we think that this is not an issue in our settings. We understand that when the analog is searched in the pool of observations and then the reconstruction is tested also against observations (e.g. as correlation to observations), the autocorrelation of the proxies would artificially boost this correlation. But in our settings the analog pool is solely provided by the model runs. The reconstructions are, in some of the tests here, compared to the observations, but since the analogues stem from model runs, and the analogues are chosen from the model pool, any correlation to observations cannot be artificially increased.

**2.3 Not clear how the AM-CFR achieves extrapolation**

The reviewer refers to an unclear use of the word 'extrapolation'. He interprets that we are referring to interpolation in time or to interpolation between chosen analogues or extrapolation beyond a certain extreme analogue. In our response in the interactive phase we indicated that:

> *"We believe this issue is rather a misunderstanding, probably caused by not using the term "extrapolation" carefully enough. We do not mean extrapolation in a temporal sense, and certainly the analogs are just pool members (or plain averages of them). Therefore it is clear that the output of the reconstruction is constrained by the variability in the pool, so*

*in this sense the method does not extrapolate anything outside the pool. Still, is worth to note that due to the multiple models merged and the length of each run, the available variability within the pool is larger than the variability observed in the calibration period.*

*In any case, what we mean when we write "extrapolation" in this context is "spatial extrapolation". It is an expression we use to refer to the general goal aimed by CFR techniques: filling spatial gaps between local reconstructions. Therefore we will re-write the parts where this expression is used to minimize chances of mislead the reader."*

We have rephrased this paragraph to avoid confusion.

**2.4   Evaluation of uncertainty**

The reviewer correctly indicates that we do not provide uncertainty ranges of the reconstructions, in the same way as they could be estimated in a real reconstruction. In our response in the interactive phase we indicated that:

*"This is an important caveat of the first version of the manuscript that has also been C6 pointed out by other reviewers. However, this question is actually much deeper than it may seem at first sight, since there is no theory -to our knowledge- to estimate es- timation uncertainty in the AM as in other established statistical methods like linear regression..."*

Indeed, we think that an estimation of the uncertainty including all potential sources of uncertainty is (still) not possible in this setting of the analogue method. Some sources of uncertainty can be estimated and we now provide an estimation of those, but the issue remains open and target for further research. We try to illustrate the problems here, although similar arguments have been developed also in the manuscript.

In the setting of a simple univariate linear regression

$$T = T_m + (P - P_m)\alpha + \epsilon,$$

with $P_m$ and $T_m$ being the mean values of $T$ and $P$, and $\alpha$ the regression coefficient. The uncertainty in the estimation of $T$ given $P$ has two main sources. One is related to the amplitude of the unresolved variance (given by the standard deviation of the error $\epsilon$). The other main source is the uncertainty in the estimation of $\alpha$. This second main contribution is roughly the product $(P - P_m)\delta(\epsilon)$. Therefore, for values of $P$ in the middle of the range of the predictor, the main contribution is the amplitude of $\epsilon$, whereas for values of $P$ far away from $P_m$, the main contribution is $\delta(\alpha)(P - P_m)$.

Analogously, in the case of the analog method we would have two main contributions. One would be the amplitude of the error term, i.e. the deviations between the predicted y and the real y assuming that the analogue is perfect. This contribution would be the unresolved variance, the variability of y that is not related to the large-scale analog. The second contribution to uncertainty in the analogue method would be the estimation of the analogue itself. However, here the situation is much more complex than in the case of regression. For targets where good analogues can be easily be found, this contribution will be very small. Since we use a vary large pool, it can be assumed that for proxy patterns that are 'around the mean' we will find good analogues in the model pool. However, for proxy patterns that are outside the range of the pool, and no good analogues can be found, the uncertainty cannot be quantified. The reason is that an analytical model, which could be the counter part of the regression model, is missing. Just by sampling it is not possible to estimate the error for analogues that are outside the range of the pool. For this purpose we would need some sort of 'extrapolation model' (here used in exactly the same sense as the reviewer in point 3). For targets well beyond the analogue pool this contribution to uncertainty would be the largest, but precisely in this range its contribution cannot be estimated, as far as we know. This situation is, to some extent, similar to pollen-based reconstructions using the analogue method. When the pollen

record shows a pattern tha is not present in the current pollen distribution, the reconstruction and its uncertainty are virtually impossible to estimate. Here, further research seems to be needed.

In this manuscript we have estimated the contribution of the 'unresolved variance'. We do opt by computing the standard deviations of the residuals (reconstructions minus target). For this computation, we put ourselves in the same situation that we would face in a real reconstruction. In this situation, we would know the observed temperature field over the last century approximately, and we therefore compute this standard deviation using only the last 150 years years of the GISS simulation (which is our target temperature). We compute this contribution of the uncertainty for two situations. One in which the proxy network is assumed complete, and one situation in which the proxy network is assumed as diminished as the real PAGES-SEL network around 1500. The corresponding results are shown in the manuscript in Figure 11.

**2.5 Indicate in Figure 1 the different nature of the proxies**

We have modified the figure to address this good suggestion.

**3 Reviewer 2**

We again thank the reviewer for the time devoted to the careful reading of the manuscript and the interesting suggestions. We elaborate here on the responses we indicated in the interactive open phase.

**3.1 I find the title and to a lesser extent the abstract to be slightly misleading**

The title and the abstract have now been slightly reworded to improve clarity. The title now reads 'Pseudo-proxy tests of the analogue method to reconstruct the spatially resolved global temperature during the Common Era'

**3.2 Equally I find the use of the acronym "MAT" slightly confusing**

We have changed the acronym and the target variable to surface air temperature SAT, and specify in each case if we refer to the global mean or the spatially resolved field.

**3.3 Reduced variance in the Arctic**

In our response we indicated that: *"Using a target infilled differently would have a minor role in the output of the reconstruction, as the analogs themselves are independent on this."* Indeed, we have to underline that the reconstruction method itself does not use observations. It only uses the proxy records and the pool of analogues from the model simulations - this is a misunderstanding also present in the review by reviewer #1, and so it shows that we did not explain clearly enough what is the input go the analogue method in this settings. We have reworded critical sentences in the manuscript to improve clarity.

The use of an alternative observations data set to estimate the skill and biases of the method could affect this estimation, but any changes would be due to the benchmark field and not to changes in the reconstructions.

**3.4 Since you discuss the sensitivity to individual models I think it would be useful to mention explicitly which models correspond to which bar in fig. 4**

In our previous response we indicated that:

*"In the first version of the manuscript we decided not to label models on purpose, since the nature of the method consist of considering all models equally as members of a pool with all models contributing equally without any consideration of their intrinsic skill, and where the method selects years blindly without other consideration than the chosen metric. Therefore we are not sure if labelling models is sensible in this context."*

We have re-considered this question and we believe that the gain in labelling the models would be little, and the clarity of the figure would be in contrast reduced. We feel this is rather a matter of perspective and we prefer to leave the figure as in the original submission, since the main objective of the study is to test the analogue method and not benchmark the individual models or compare the model simulations to each other, for which other studies have and will provide a more detailed picture.

**3.5 The majority of the results presented are obtained using only one model (GISS r1i1p121) as the target**

We have indeed carried out similar analysis using other model - the MPI-ESM model as target, albeit we have not repeated the whole analysis using one target model at a time. The results obtained with the MPI model as target as closely similar to those obtain with the GISS model, and this is now indicated in the manuscript.

**3.6 In the theory or discussion section no mention is mused to provide uncertainties to the reconstructions**

This very same issue is discussed in Section 2.4 in this document.

**3.7 No mention is also made regarding the drop in coverage of the proxy network back through time**

This has been due to a possibly unclear description of the experiments. The pseudo-proxies are generated with realistic dates of the missing values, so that the pseudo-proxies networks do mimic the sparser networks of real proxies back in time.

**3.8 In section 5 you comment that the difference between figure 7 and figure 5 6 suggests that the level of noise employed in the first PPE is an underestimation. Could it not also be due to errors in the model fields?**

This explanation is now included in the discussion. This is also a possibility that is also included in the discussion of this figure. Thank you very much for this suggestion.

**3.9 If one of the goals of this paper is to lay the ground work for a future analysis which reconstructs the global mean climate of the last 1000 or 2000 years then I think that a comparison of your global annual mean reconstruction (fig 9) with simpler commonly used reconstructions methods which only use the proxies scaled to the observations**

The reviewer is referring to a comparison of the analog method to the application of Composite plus Scaling (CPS), and in our previous response we agreed that this would be an interesting comparison. However, after more careful considerations and taking into account the expansion of the manuscript

to include the uncertainties, we have now second thoughts. First, CPS provides just the global mean temperature, whereas one of the clear strength of the Analogue Methods is the reconstruction of the spatially resolved fields. This comparison would therefore be rather limited and would suggest that the Analogue Method just reconstructs the global mean as well. The alternative, namely to compare the Analogue Method to other spatially field reconstructions methods, such as methods based on Principal Component Regression, Regularized Expectation Maximization or Bayesian Hierarchical Modelling would indeed expand the manuscript to an unacceptable length, even more so since each of these methods have their own variants.

Therefore we leave it to the decision of the editor whether a limited comparison solely to CPS would be meaningful in this manuscript, or rather whether a forthcoming study focused solely on a comparison of the Analogue Method to several climate field reconstruction methods would make more sense. **We would appreciate the feedback of the editor on this point.**

**3.10  Minor copy-edits**

Thank you for the careful reading. They have been corrected.

[revised manuscript text omitted]

---

## Author Response (AR3)

**Point by point authors response to second round of review**

J. J. Gómez-Navarro on behalf of all co-authors

May 4, 2017

*Generally, I find this paper greatly improved and would recommend that it be published provided that some mainly minor revisions are performed.*

We thank the reviewer for carefully reading again the manuscript and providing more useful insight that will surely improve the final version of the manuscript.

*It is good that the authors take into account the change in the proxy network back in time and this is now clearer in the text but I feel this point should be expanded on when describing the performance of the reconstruction. The period in which the correlations to the true values etc are evaluated will presumably impact the results due to the difference in coverage through time. In particular, when comparing the correlations in the figure 8 to figures 5,6,7 are they calculated over the same time period? If not wouldn't the difference in proxy coverage effect the results, making any comparison problematic?*

It is true that the correlations can not be directly compared. We have added sentences (highlighted in blue in the tracked-chages document) that emphasise this important point, initially overseen by us.

*Also in the figures, is the proxy network plotted showing maximum availability? If so I think it would be useful to state that some of these proxies do not cover the full period over which the correlations are calculated and perhaps the location of the proxies which do cover the full period could be indicated.*

We have added a new map as part of Figure 1 that illustrates the availability of proxy data. Note however it is beyond the scope of this manuscript to thoughtfully review the dataset. Therefore we refer explicitly to the original paper (Open Access) for further information of the temporal evolution of proxy availability.

*Somewhere in the paper it should be mentioned that the reconstructions show less homogeneity back through time than the models (e.g. co-variability between the NH and SH) see Neukom et al and the paper by the PAGES 2k-PMIP3 group (Continental-scale temperature variability in PMIP3 simulations and PAGES 2k regional temperature reconstructions over the past millennium). This could mean that the performance of the PPE examples, particularly in the SH, shown here may be optimistic when compared to the performance in a real world scenario.*

This is indeed an interesting observation. We have included a whole new paragraph in the Conclusions section that acknowledges this issue.

*Figure 4 – As mentioned in my previous review I really think this figure would illustrate the point you wish to make far better if you make panels b and d a perfect square and put a x=y line through it. This would make it much more intuitive and clearer to interpret. In addition I think marking where the major volcanic eruptions occurred would help. At the moment I am struggling to identify for example whether following 1258, 1815, volcanic years in the models are preferentially selected for analogs.*

The figure has been modified according to the reviewer suggestion.

*In section 7 it is not clear to me why you have used the case with uniform white noise instead of the more realistic proxy noise setup, can you justify this? I would have thought that the most realistic set-up would be a better choice.*

The reviewer is right, so we have replaced the figure showing the most realistic case, although it supports the same conclusions.

*Abstract. Assuming AM stands for analogue method, the definition should be moved to line 8 for clarity. "variants of the Analog Method (AM) . . . " And any mention of the "analog method" elsewhere in the text should be replaced with AM, please check the introduction especially.*

This convention has been implemented across the whole manuscript.

*Line 35 "to finally obtain a complete"*
*Line 41 "each gird-cell of"*
*line 70 and 72 – check brackets around references.*

These changes have been implemented

*Line 179 – do you mean ensemble median? This is the most commonly used HadCRUT4 product. . .*

The reviewer is right, so we have edited the text accordingly.

*Line 229 – the forcings are similar but not totally consistent i.e. different volcanic and solar forcing datasets are used. This should be mentioned and a citation to Schmidt et al 2012 added.*
*Line 244 – "HadCRUT4 dataset"*
*Lines 369-373 – Could a sentence be added here expanding why using the GISS model which has a lower variance would result in a stricter test. This wasn't initially clear to me.*
*Line 417 "has the effect"*
*Line 473 "the effect of non-climate related"*

Various minor text edits have been implemented to address all these issues.

*RProxy PPE section – I would like to see a slightly expanded discussion of the degradation of results. To me this looks to be particularly prominent in the tropics where the proxy data has a poorer correlation to the observations. Is this correct?*

We have enlarged such discussion. However, this time we believe it is not easy demonstrate the guess of the reviewer. It is true that the differences are stronger in the tropics, but the reason seems to be that correlation was originally very high there, rather than because the skill is worse in relative terms (i.e. in locations where the skill was already low, the degradation is obviously lower). Therefore, we believe the performance degradation is more related to lose of absolute skill in areas where it is relatively large, and not so clearly related to the nature of proxies in those locations. Actually, in the tropics the correlations are rather high (see dark blue squares in Fig. 1). As the argument seems not very strong, we have decided not to include it in the discussion.

*Line 605 – The observations in the high Arctic are based on in-filled data and not real observations. Therefore the comparison of variance in models and observations in this region should be caveated.*

This is an interesting and fair point that we have included in the discussion of the result.

*Section 6. Could the improvement in performance of the screened network also be due to removing the proxies with poorer correlations with observations from the network?*

This is actually the case. We have stressed this point in the discussion.

*Line 675 – This sentence is not clear.*

We believe the problem of this sentence is that it heavily relies on the context provided by the former paragraph. We have tried to rephrase it to make it more clear.

*Figure 7 – Why has the variance preservation plot log($\sigma/\sigma$) not been shown as in figures 2,3,5,6? I think this would be useful to include.*

We wanted to include a measure of RMSE to illustrate other metrics, as suggested in the first stage of the review. Therefore we decided to remove the results for variability to keep a rather small figure. However, it might have been a bad idea, as pointed out by the reviewer. Therefore we now include both RMSE and standard deviation, and we briefly discuss all results together from the three metrics shown in the figure.

*Figure 11. Finding that the poles give the largest error is perhaps not that surprising given that this is the part of the world with the most variability. Could you add plots showing the errors divided by the models control variability in each grid cell as I think this would give a better demonstration of the performance of the AM.*

We have carried out the analysis, and Figure 11 shows now the suggested calculations.

[revised manuscript text omitted]